# The Off-Label Use of Selective Serotonin Reuptake Inhibitors for Sexual Behavior Management: Risks and Considerations

**DOI:** 10.3390/healthcare13192433

**Published:** 2025-09-25

**Authors:** Jonathan Shaw, Charles Lai, Peter Bota, Andrew Le, Anton Andricioaei, Theodore Tran, Tina Allee

**Affiliations:** 1School of Medicine, California University of Science and Medicine, Colton, CA 92324, USA; 2Department of Chemistry and Physics, Drury University, Springfield, MO 65802, USA; 3Department of Anthropology, University of California, Riverside, CA 92521, USA; 4Psychiatry, College Medical Center, Long Beach, CA 90806, USA

**Keywords:** hypersexuality, SSRI, medication management, psychiatry, impulsivity

## Abstract

Background: Selective serotonin reuptake inhibitors (SSRIs) are one of the most frequently used medication classes in psychiatry, with many approved and off-label uses. One common side effect of SSRIs is sexual dysfunction, leading to the off-label use of SSRIs to manage inappropriate sexual behaviors in psychiatric settings. However, no official guidelines exist for this off-label use of SSRIs, so a review of this use is warranted. Methods: This review was conducted using the PubMed and Google Scholar databases. Grey literature was considered for inclusion in this review, but only one report by the United Kingdom’s Care Quality Commission was included. Peer-reviewed references discussing the theoretical mechanisms of SSRI-induced sexual dysfunction, case reports/studies examining the off-label use of SSRIs, and reviews discussing relevant disorders like post-SSRI sexual dysfunction (PSSD) were included in this review. Results: The literature proposes that SSRIs act through a variety of serotonin receptors such as 5-HT_1A_, 5-HT_2A_, and 5-HT_2C_ to inhibit dopaminergic tone in the mesolimbic and spinal pathways to cause sexual dysfunction. Discussion: SSRIs are frequently considered for off-label use in managing inappropriate sexual behavior, particularly in geriatric patients with dementia, given their superior safety profile compared to antipsychotics in that population. However, the risk and treatment options for PSSD are unclear, which poses a risk for patients taking SRRIs, as it can be a severe and enduring condition. High-quality clinical trials are needed, as the majority of the literature on the topic consists of case reports or theoretical papers.

## 1. Introduction

Inappropriate sexual behaviors, defined as physical or verbal actions with sexual content and/or intent outside of the established social norms of a situation up to and including coercive and unwanted sexual advances, can present in psychiatric or neurologic patients and disrupt the therapeutic relationship between the patient and provider. These inappropriate sexual behaviors can pose a potential threat to psychiatric staff, particularly those in frequent contact with patients such as sitters and nursing staff. An analysis of nearly 60,000 reports between April and June of 2017 filed with England’s National Health Service Trust regarding mental health wards found that 1120 (1.6% of reports) were sexual incidents involving patients, staff, and visitors [1]. About two-thirds (594) of the people affected were categorized as patients and one third (301) were staff, with a small number (24) who were others, such as visitors to the ward [1]. Therefore, the management of inappropriate sexual behaviors is paramount for maintaining not only staff safety but also the safety of other patients.

The causes of sexually inappropriate behavior can be understood as either sexual disinhibition or hypersexuality. Sexual disinhibition presents as inappropriate sexual acts that stem from inadequate self-control. Similarly, hypersexuality presents as repeated and intense sexual urges or fixations that are difficult to control and can therefore cause distress and inappropriate behavior in those affected. Regardless of the cause of patient sexual disinhibition or hypersexuality, inappropriate sexual behaviors pose a potentially significant risk to staff in an inpatient psychiatric setting with no readily available standardized guidelines for addressing incidents before they occur [1].

Although non-pharmacological strategies, such as verbal de-escalation, remain foundational, clinicians frequently turn to pharmaceutical interventions utilizing medications with sedating or libido-reducing properties despite a lack of existing clinical guidelines for this indication [1]. Selective serotonin reuptake inhibitors (SSRIs) are known to frequently cause sexual dysfunction as an adverse effect and are therefore used off-label for inappropriate sexual behaviors [2,3]. SSRIs were developed in the late 1970s with the serendipitous discovery of fluoxetine, which later received Food and Drug Administration (FDA) approval in 1987 [4]. Primarily used for the treatment of major depressive disorder (MDD), SSRIs have greater receptor selectivity and superior safety profiles compared to older treatment options for depression such as monoamine-oxidase inhibitors and tricyclic antidepressants [3,4,5]. By 2001, SSRIs accounted for nearly 70% of all antidepressant prescriptions in the United States [6].

Given their widespread adoption and superior safety and tolerability profile versus prior antidepressants, SSRIs have been explored as a treatment modality for various disorders and purposes, including being an FDA-approved first-line treatment for general anxiety disorder, panic disorder, social anxiety disorder, post-traumatic stress disorder, obsessive–compulsive disorder, premenstrual dysphoric disorder, and bulimia nervosa [3,7]. The broad utility of SSRIs in psychiatry has led to the exploration of off-label applications such as the management of inappropriate sexual behaviors [3,5]. However, clinicians must always balance the benefits of an intervention against its risks. Although SSRIs are generally well tolerated, there are still common adverse effects including sexual dysfunction, gastrointestinal upset, insomnia, and weight change [3,8]. Patients may also experience QT-interval prolongation or hyponatremia, mediated by the syndrome of inappropriate antidiuretic hormone secretion (SIADH), which can be life-threatening and may occur in up to 4% of hospitalized patients taking SSRIs [9]. Given the potential for life-threatening complications and the expanding body of literature regarding post-SSRI sexual dysfunction (PSSD), a critical analysis of the risks and considerations surrounding off-label SSRI use for managing inappropriate sexual behaviors in psychiatric settings is warranted.

In this narrative review, the proposed mechanisms of the negative sexual side effects associated with SSRIs, which affect 25–88.7% of patients treated with SSRIs, will be discussed in the context of harnessing this side effect to manage inappropriate sexual behaviors in psychiatric settings [10,11]. The serotonergic stimulation of 5-HT_2C_ receptors has been proposed to inhibit dopaminergic tone within the mesolimbic and spinal pathways, resulting in long-lasting PSSD in between 0.46% (≈1 in 216 treated individuals) and 26.3% of patients treated [12,13,14]. PSSD, which presents with similar proposed mechanisms of action and symptoms to a disorder called post-Finasteride syndrome (PFS), is a poorly understood, underdiagnosed, and undertreated disorder with limited clinician awareness and a lack of standardized diagnostic criteria [15]. Potentially long-term and poorly understood side effects like PSSD carry serious ethical implications for the use of SSRIs for the off-label purpose of managing inappropriate sexual behaviors; however, the findings of prior systematic reviews endorse a meaningful reduction or resolution of inappropriate sexual behaviors in 72% of geriatric psychiatry patients, an efficacy which merits further investigation and study [16].

Additionally, this narrative review will also evaluate the strength of the evidence for the off-label use of SSRIs in managing inappropriate sexual behaviors as, while there is currently a lack of clinical trials on this subject, several individual case reports document relatively rapid cessation of inappropriate sexual behaviors following the initiation of citalopram or paroxetine in nursing-home residents with dementia [3]. This evidence and the associated risks and benefits of SSRIs in this particular off-label use will be compared against other commonly used pharmaceutical options for reducing inappropriate sexual behaviors, such as anti-androgens or anti-dopaminergic medications. The present review will provide clinicians with suitable information to provide balanced, patient-centered clinical decision making when considering the off-label use of SSRIs for managing inappropriate sexual behaviors in inpatient psychiatric settings.

## 2. Methods

This is a narrative review examining the efficacy and history of the off-label use of SSRIs for managing inappropriate sexual behavior among patients in inpatient psychiatric units. No formal protocol was registered as this is not a systematic review, but the following guidelines and criteria were employed. 

### 2.1. Reference Search Strategy

The authors primarily used Google Scholar and PubMed in order to identify relevant references. An initial search with the query, “Selective serotonin reuptake inhibitor clinical trials for managing sexual behaviors,” yielded 22,800 results on Google Scholar and 34 on PubMed. The authors reviewed the titles and abstracts of the 34 PubMed references, finding clinical trials examining the use of SSRIs for managing sexual behaviors in psychiatric patients. As 22,800 results was deemed too broad of a search, the Google Scholar query was further refined using Boolean operators to search, “(Selective serotonin reuptake inhibitor) AND (clinical trial) AND (managing sexual behaviors),” alongside including criteria for references to be published from 2021 onward. This resulted in 16,800 results. After sorting be relevance instead of publication date, the authors reviewed the titles of the first 500 references and found no clinical trials explicitly examining the use of SSRIs for managing sexual behavior, instead the vast majority of the references discussed off-label uses of SSRIs (such as its use for premature ejaculation) or its sexual side effects in general.

At this point, the authors had concluded there are likely no peer-reviewed publications of clinical trials examining the use of SSRIs for managing sexual behaviors in psychiatric patients. Instead, the authors used PubMed in order to identify clinical trials and theoretical papers which discuss the prevalence of SSRI induced sexual dysfunction, how to treat SSRI induced sexual dysfunction, and the proposed mechanisms for SSRI induced sexual dysfunction. The authors queried, “(Selective serotonin reuptake inhibitor) AND (sexual dysfunction) AND (…)” with the last search term being replaced with the keywords: prevalence, treatment, and mechanism. This yielded 235 results, 151 (restricted to published within the past 5 years) results, and 167 results, respectively. The authors then screened the titles and abstracts of these references for inclusion in this review. The authors also input the titles of relevant references into Google Scholar to see if these references were cited by more recent peer-reviewed publications that would also be suitable for inclusion in the review. This led to further PubMed searches regarding the prevalence and treatment of PSSD, the proposed mechanism of PFS, and the benefits and risks of anti-androgen as well as anti-dopaminergic medications for managing inappropriate sexual behaviors.

### 2.2. Study Eligibility

To be included in this review, an English version of the manuscript must have been available to the authors and the reference must have been a peer-reviewed publication in an academic journal. The authors primarily sought clinical trials but also included case reports as points of evidence. References discussing proposed mechanisms were also included. For the most part, grey literature was not included in this narrative review. However, the authors did examine grey literature as a way to better direct database searches into relevant topics related to SSRI use for sexual behavior management. This included viewing the list of references on the website of an organization called the, “PSSD Network,” which is a non-profit charity organization based in Australia, led by PSSD patients and their loved ones. The only reference included that would be considered grey literature is the 2018 report by the United Kingdom’s Care Quality Commission that examined incidents related to sexual safety in National Health Service mental health trusts as a source of epidemiological data.

### 2.3. Reference Bias Assessment

The risk of biases was assessed using the JBI critical appraisal checklist for case reports, the SYRCLE’s risk assessment tool for animal studies, the Scale for the Assessment of Narrative Review Articles, and the National Institutes of Health’s study quality assessment tools for case series studies as well as observational cohort and cross-sectional studies [17,18,19,20,21]. The results of these bias assessments can be seen in Appendix A. Authors J.S., C.L., and P.B. assessed each reference as a pair. If a disagreement on a bias rating question occurred, then the third coauthor would act as a tiebreaker.

## 3. Results

### 3.1. Proposed Mechanisms

As indicated by its name, a selective serotonin reuptake inhibitor’s mechanism is the enhancement of serotonin (5-HT) signaling by inhibiting its reuptake at synapses, leading to increased synaptic availability [22,23,24]. This mechanism is associated with delayed orgasm, decreased libido, and erectile or lubrication difficulties, effects that have been consistently documented across diverse populations [2,11,25,26,27,28,29,30]. These same properties are therapeutically leveraged in cases of sexually inappropriate behaviors in psychiatric settings, as increased serotonergic tone is associated with reduced sexual drive and impulsivity [3,31,32,33,34].

One of the central pathways involved in sexual arousal and reward is the mesolimbic dopaminergic system [31,35,36,37,38,39]. SSRIs indirectly suppress this system by increasing serotonin in areas such as the nucleus accumbens and prefrontal cortex [15,36,40]. Elevated serotonin inhibits dopamine release, blunting the motivational and reward-related components of sexual behavior [15,39,41,42,43]. This is particularly relevant in treating patients exhibiting hypersexual or compulsive sexual behaviors, where dopamine-mediated reinforcement loops may drive problematic actions [35,38,43,44,45]. However, the onset of sexual side effects parallels that of the antidepressant activities of SSRIs, typically developing after chronic administration over the timeframe of weeks instead of acute administration of SSRIs [39]. This suggests that chronic SSRI treatment affects sexual behaviors by enhancing 5-HT activity in areas of the brain, such as the mesolimbic pathway, where 5-HT can influence the release of dopamine [39].

Additionally, long-term or persistent sexual side effects, even after discontinuation of SSRIs, suggest deeper neuroplastic or neuroendocrine changes [13,15,36,46]. PSSD points to long-term downregulation or desensitization of receptors involved in arousal and sexual response, but this syndrome also has broader implications, such as alterations in emotional functioning [14,22,27,40,47,48,49,50,51]. These effects provide indirect evidence that SSRIs exert their behavioral suppression by remodeling neural sensitivity to sexual stimuli, reinforcing their potential therapeutic role in conditions involving inappropriate sexual behaviors [36,38,41,52,53].

SSRIs can act on a variety of serotonin receptor subtypes, but emerging research indicates that the 5-HT_2A_ and 5-HT_1A_ receptors play pivotal roles in sexual regulation. Activation of 5-HT_2A_ receptors has been associated with decreased sexual arousal, while antagonism of 5-HT_1A_ receptors reduces sexual compulsivity [15,22,54]. Of note, excitation of the 5-HT_1A_ receptor by an agonist is associated with pro-sexual effects, such as an improvement of the sexual side effects associated with SSRIs, which partially explains the differences in sexual side effect profiles between various antidepressants [55,56]. This was supported through preclinical serotonin transporter knockout (SERT−/−) rat-models which exhibited enhanced extracellular 5-HT levels and desensitized 5-HT_1A_ receptors [55]. Male rats lacking the serotonin transporter (SERT−/−) exhibit a nine-fold increase in extracellular 5-HT levels as well as lower basal ejaculatory performance than wildtype rats (SERT+/+) or heterozygous serotonin transporter knockout (SERT+/−) rats [55]. Additionally, augmentation of SSRIs with mirtazapine seems to decrease the SSRI-associated sexual side effects, supporting the theory that SSRI-induced sexual dysfunction may be mediated by 5-HT_2C_-receptors [39]. 23,2006, a selective 5-HT_2C_-receptor antagonist, demonstrates antidepressant effects in preclinical depression models and elevates extracellular dopamine and norepinephrine, but not 5-HT which suggests that 5-HT_2C_-receptors are potentially involved with SSRI-induced sexual side effects [39]. However, the existing literature that examines the roles that the various 5-HT receptor subtypes play in SSRI-induced sexual dysfunction is largely preclinical, mainly with rat-models, so further research is needed to better illustrate the exact mechanisms of SSRI-induced sexual dysfunction. By understanding these subtype-specific actions, clinicians can make well-informed clinical decisions based on the differential effects of various SSRIs and this will support targeted therapeutic approaches in managing inappropriate sexual behaviors or in managing sexual side effects that occur while using SSRIs as an antidepressant.

Sexual behavior involves a balance of impulse control, reward processing, and executive function [34,35,36,57,58]. SSRIs enhance top-down inhibitory control by modulating prefrontal-limbic circuits [40]. This improved inhibitory function is particularly valuable in psychiatric populations where inappropriate sexual behaviors may arise from impaired judgment or poor emotional regulation [57]. Enhancing serotonin transmission in the orbitofrontal cortex and anterior cingulate appears to increase behavioral inhibition, leading to reductions in inappropriate or compulsive sexual behaviors, albeit not as significantly as anti-androgens [25]. This was observed in incarcerated individuals with a history of sexual offenses where SSRIs were seen to reduce sexual preoccupation and fantasies [25]. The mechanism is presumed to be multifactorial: serotonergic suppression of libido, dampened dopaminergic drive, and reduced sexual ideation due to frontal lobe modulation [25,42,43]. 

SSRIs may alter levels of neuroactive steroids such as testosterone and estrogen by affecting the hypothalamic-pituitary-gonadal axis [15,22,31,36,44]. However human trials examining this mechanism are limited, with those which have been conducted not providing publicly available data like with NCT00611975. However, there have been multiple animal and bench studies that have been published examining this proposed mechanism. Studies using proteasome-wide in silico screening and patient surveys have suggested that SSRIs influence steroidogenesis, with downstream reductions in androgens contributing to lowered sexual interest [47,59]. In male rats, SSRIs like fluoxetine were found to decrease the activity of 17-beta-hydroxysteroid dehydrogenase type 6, an enzyme involved in steroidogenesis, which inhibits the steroidogenic capacity of Leydig cells and could contribute to the decreased serum concentrations of luteinizing hormone, follicle stimulating hormone, progesterone, and testosterone seen in male rats treated with SSRIs [60]. Meanwhile in prepubertal female rats, SSRIs like fluoxetine are thought to affect follicular development and the ovulation process by acting on the hypothalamus-pituitary-gonadal axis by increasing estradiol concentrations with acute administration and increasing serotonin concentrations in ovaries in subchronic administration [60]. These endocrine changes may contribute to the therapeutic sexual dampening seen in both men and women treated for hypersexuality or paraphilic disorders [25,46,48,61].

SSRI use during adolescent neurodevelopment has been linked to altered adult sexual behavior and function, potentially due to long-lasting changes in serotonergic and dopaminergic systems [57]. While this raises concerns in therapeutic contexts, it also underscores the enduring impact SSRIs can have on shaping sexual behavior, which could be utilized therapeutically in carefully controlled settings. Another important consideration in long-term SSRI use is tolerance to sexual side effects, which may diminish over time or be addressed by the concept of “drug holidays,” in which SSRIs are temporarily discontinued to allow sexual function to recover [31]. This strategy is used in patients where the therapeutic benefit of SSRIs in controlling sexual behavior needs to be balanced with the patient’s desire for preserved sexual function, and findings suggest the sexual-suppressive mechanisms are reversible to some extent and modifiable depending on treatment goals [31].

Not all SSRIs exert identical effects on sexual behavior; some agents, such as paroxetine, demonstrate relatively stronger suppressive effects on sexual behaviors due to their antidopaminergic properties and higher affinity for serotonin transporters, whereas others like fluoxetine may have milder impacts [11,31,54]. Other antidepressants, such as the serotonin antagonist and reuptake inhibitor trazodone, present with lower rates of sexual dysfunction compared to SSRI antidepressants, potentially due to its dual mechanism of action involving inhibition of SERT and serotonin type 2 receptor antagonism (both the 5-HT_2A_ and 5-HT_2C_ receptors) [56]. Preclinical evidence suggests that antagonism of SERT results in serotonin exerting its agonistic actions on the 5-HT_1A_ receptor which exerts pro-sexual effects, potentially contributing to the lower prevalence of sexual side effects associated with trazodone [55,56]. Understanding these differences allows clinicians to tailor treatment based on the severity of sexual symptoms and the therapeutic goal, whether to reduce sexual drive in inappropriate behaviors or to minimize side effects in depressed patients. The odds ratios of sexual dysfunction associated with various antidepressants can be seen in Table 1 below.

### 3.2. Post SSRI Sexual Dysfunction and Similar Disorders

PSSD is increasingly being recognized as a persistent condition characterized by sexual side effects that continue even after discontinuation of SSRIs [15,22,27,48]. While the overall prevalence is difficult to establish due to underreporting, variability in study designs, and the historical lack of formal diagnostic criteria, a growing body of literature suggests that it may affect between 0.46% (1 in 216) to 26.3% of SSRI users [13,14,22,29]. This wide range in reported prevalence is potentially due to the lack of standardized diagnostic criteria which can easily result in under or over-reporting of the incidence of PSSD [14,15]. A large-scale systematic review found that a significant number of patients experience enduring sexual dysfunction following the cessation of SSRIs, including symptoms such as genital numbness, anorgasmia, erectile dysfunction, and decreased libido [22]. However, this systematic review concludes that no accurate prevalence for PSSD can be inferred from the existing data as it is mainly from case reports and online surveys [22]. Despite being unable to accurately quantify the prevalence of PSSD, the severity of this side effect is still important given that this persistent sexual dysfunction may last for months or even years, and in some cases, appear irreversible [22,24]. Some studies have found that sexual dysfunction is highly prevalent among both females (88.7%) and males (84.5%) taking SSRIs or SNRIs for psychiatric conditions [11]. Despite this prevalence, concerns exist about underreporting and lack of recognition of these symptoms from clinicians [13,23,29,31].

Another condition exists, known as PFS, with remarkably similar symptoms such as persistent sexual, cognitive, and emotional side effects following the use of 5α-reductase inhibitors like finasteride [46,47,48,50]. Studies in the literature have found striking similarities between PSSD and PFS in terms of symptom profiles, neurosteroid dysfunction, and patient-reported distress [46,48,50]. In both syndromes, patients often report a sudden onset of dysfunction that persists beyond the expected pharmacological window, suggesting underlying neurobiological or hormonal alterations [46,48,50]. It has been demonstrated in adult male rate-models that after 20 days of finasteride treatment and its withdrawal (i.e., one month of suspension), levels of neuroactive steroids in the plasma, cerebrospinal fluid, and central nervous system areas (cerebral cortex, cerebellum and hippocampus) are affected [62]. During finasteride treatment, progesterone levels were increased in the hippocampus but not in the cerebral cortex and cerebellum while levels of isopregnanolone and 3α-diol were increased and decreased, respectively, in the cerebellum; these changes did not occur in the cerebral cortex and hippocampus [62]. After withdrawal of finasteride, certain changes like dihydroprogesterone levels remaining elevated in the cerebellum stayed the same, while new changes that did not occur during treatment such as an increase and a decrease in pregnenolone levels in the cerebellum and hippocampus, respectively, and a decrease in allopregnanolone in the cerebral cortex [62]. These changes to the levels of neuroactive steroids, their precursors, and even an increased expression in androgen receptors in the cerebral cortex during and after treatment with finasteride results in the clinical presentation of PFS: lower orgasmic function, sexual desire and overall satisfaction domains compared to the general population [62,63]. 

Although the estimated prevalence of PFS remains low due to limited awareness and diagnostic ambiguity, its increasing documentation in medical literature offers a parallel lens through which to view PSSD [48]. A proteome-wide in silico study identified molecular targets potentially involved in the pathophysiology of PSSD and PFS that led to disruptions to neurosteroidogenic enzymes and sex hormone pathways, particularly those affecting androgen receptors and estrogen balance [47]. This biological overlap reinforces the notion that these post-drug syndromes may not be psychosomatic but rather reflect structural or receptor-level changes induced by pharmacotherapy [47].

Despite mounting evidence, the true prevalence of PSSD remains uncertain due to a lack of longitudinal studies and standardized reporting systems [24,29,53]. Many cases go undocumented, and patients may be misdiagnosed with psychogenic sexual dysfunction or relapse of mood disorders [29,53]. Nonetheless, anecdotal data and surveys suggest that a small but significant proportion, reportedly between 0.46% and 26.3% of SSRI users, may experience persistent symptoms post-treatment [13,14,22,48]. Given the parallels with PFS and the distressing, often debilitating nature of these syndromes, increased clinical awareness and systematic investigation are urgently needed. As seen in Figure 1, both PSSD and PFS challenge the traditional view of psychotropic and endocrine-modulating medications as reversible in effect and demand a rethinking of long-term risk–benefit analyses in prescribing practices.

### 3.3. Hypersexuality Associated with SSRI Use

While SSRIs are primarily associated with sexual dysfunction, there are rare but noteworthy reports of paradoxical hypersexuality during treatment [35,38,43]. However, the overwhelming majority of patients experience reduced sexual function, highlighting a stark contrast in frequency and clinical presentation [11]. Sexual dysfunction is among the most common adverse effects of SSRI therapy, with prevalence estimates ranging from 84.5% to 93% depending on the population and specific medication used [11]. These dysfunctions include decreased libido, delayed orgasm, anorgasmia, erectile difficulties, and genital anesthesia, which is a similar presentation to PSSD, but by comparing the timeline of symptoms, one can see that SSRI-induced sexual dysfunction occurs during treatment with SSRIs while PSSD occurs after discontinuation of SSRIs [15,40,54]. These effects are dose-dependent and often underreported due to the lack of routine screening in psychiatric settings [11,22]. In contrast, reports of SSRI-induced hypersexuality are far less frequent and primarily limited to isolated case studies [35,38,43]. Of note, the hypersexuality experienced by patients in two out of three of the case reports were attributed to sertraline while one was attributed to escitalopram [35,38,43].

In the case of the 55-year-old married Caucasian male, the patient presented to management of major depressive disorder and post-traumatic stress disorder [43]. The patient was already being treated with bupropion extended release 100 mg daily, but this was inadequate for managing his symptoms, so sertraline was added to his medication regimen and titrated to 100 mg daily [43]. Upon follow up, the patient reported improved management of his symptoms but also reported increased sexual desire and demands for sexual intercourse which resorted in marital discord [43]. The patient and his wife believed that this increase in sexual desire was attributable to the addition of sertraline to the patient’s medication regimen as there were no reported sexual symptoms while on bupropion alone [43]. All other potential causes for these sexual symptoms were ruled out and the patient was discontinued on sertraline with a gradual resolution of hypersexuality over the course of one month without concomitant worsening in the symptoms of his depression or post-traumatic stress disorder [43]. The authors of this case report were unable to determine if the hypersexuality experienced by the patient was a result of sertraline alone or through a synergistic effect between sertraline and bupropion, noting that sertraline is not an exclusively serotonergic agent but also acts on norepinephrine and dopamine receptors [43]. Similarly, fluoxetine has been shown to increase synaptic dopamine levels which the authors speculate that hypersexuality can present as a separate side effect from the pro-dopaminergic effects of some SSRIs [43].

In the case of the 42-year-old married Caucasian male with a history of papillary thyroid carcinoma and no prior psychiatric history, the patient was started on bupropion extended release 150 mg once daily for moderate depression, but this was stopped due to increased irritability [38]. The patient was then started on sertraline 50 mg once daily, resulting in not only significant improvements in his depressive symptoms, but also adverse effects that were hypersexual (increased sex drive, constant sexual thoughts, and compulsive masturbating throughout the day, including strong urges to do so at work) and hyposexual (delayed ejaculation) [38]. Due to these sexual symptoms, the patient was switched to duloxetine 30 mg once daily and then escitalopram, neither of these medications were effective for his cognitive symptoms but they also did not cause any sexual side effects either [38]. As the benefits of sertraline were perceived to outweigh the risks of the patient’s sexual adverse side effects, the patient was restarted on sertraline 50 mg once daily, however he experienced no symptoms of hypersexuality, and delayed ejaculation was subjectively less severe than with the prior trial [38]. The authors of this case report comment on the potential drug–drug interaction between bupropion and sertraline of the previous case report, stating that sertraline potentially inhibits the metabolism of bupropion through the inhibition of CYP2B6 [38].

In the case of the 25-year-old married female who presented with severe depression without psychotic symptoms, the patient was started on escitalopram 5 mg daily which was then titrated up to 10 mg daily within a week with improvements in her symptoms, though not complete resolution [35]. After two months, the dose of escitalopram was increased to 15 mg per day and the patient began to experience increased sexual drive and preoccupation with sexual thoughts, such as the desire to masturbate throughout the day which she acted upon when alone, within five days [35]. The patient also began to request for more sexual activity from her husband who she was having marital conflicts with after his extramarital affair, resulting in the patient feeling distress [35]. All other potential causes of this hypersexuality were ruled out and her escitalopram was stopped, instead the patient was started on mirtazapine and oral benzodiazepines to treat her depression and anxiety [35]. The authors of this case report also believe that the probable mechanism underlying SSRI-induced hypersexuality can be attributed to serotonin receptors 5-HT_2_ and 5-HT_3_, dopamine and prolactin, and nitric oxide synthetase [35].

Some studies suggest that, in rare instances, serotonergic modulation might disinhibit sexual behaviors in susceptible individuals, particularly during developmental exposure or early treatment phases [57]. However, the frequency of such reactions remains poorly quantified, and most published literature emphasizes suppression rather than activation of sexual drive [25,57]. Despite the limited data, there appears to be a potential correlation between the concomitant use of bupropion and SSRIs such as fluoxetine or sertraline and the occurrence of hypersexuality as an adverse side effect [35,38,43]. It is generally agreed upon that hypersexuality is a dose-dependent side effect for SSRIs, so clinicians should carefully consider the potential for harm if a patient were to experience hypersexuality as a side effect from augmenting their current treatment regimen with SSRIs or were to be started on an SSRI that have been previously associated with the side effect [35,38,43]. These SSRIs include fluoxetine, paroxetine, fluvoxamine, citalopram, and escitalopram [38]. Overall, sexual dysfunction is a predictable and frequent side effect of SSRIs, whereas hypersexuality is a rare and poorly understood phenomenon. Current evidence strongly supports the former as a common pharmacological outcome, while the latter appears idiosyncratic and exceptional in clinical practice [11,22,57].

### 3.4. Comparison to Other Pharmaceutical Options

Of the potential pharmaceutical interventions which can be employed for the management of inappropriate sexual behaviors, SSRIs have been used alongside anti-dopaminergic medications (also known as antipsychotics) as well as anti-androgenic medications. To make informed clinical decisions regarding which class of medicine to use for managing sexually inappropriate behaviors in psychiatric patients, the risks and benefits of these classes must be considered. This is especially important given the lack of guidelines and FDA approvals for all of the aforementioned drug classes, meaning the use of these medications would be considered off-label.

Regarding the safety of anti-dopaminergic medications, the use of this drug class has been explored for the management of inappropriate sexual behaviors in the elderly, especially in patients with dementia [3]. Antipsychotics are frequently used to treat the behavioral and psychotic symptoms associated with dementia, however there are two black box warnings about the use of first and second-generation antipsychotics in geriatric dementia patients due to the potential for significant and potentially lethal side effects [3,5]. There is a small, but definite increase in the risk of stroke or cardiac events in patients with dementia who are treated with antipsychotics which has led to an avoidance of their use in dementia patients where possible [3,5]. More common side effects are sedation and impaired cognitive function that occur in both first and second-generation antipsychotics, though this depends on the receptor profile of the specific antipsychotic being used [3]. First-generation antipsychotics are associated with a higher risk of anticholinergic effects and can also cause dermatological, hematological, or endocrine problems albeit more rarely [3]. Second-generation antipsychotics have a lower risk of extrapyramidal symptoms compared to their first-generation counterparts but have a higher risk of causing weight gain or metabolic syndrome which can be important factors to consider given the prevalence of heart disease and diabetes in the geriatric population [3].

Some researchers have also posited the idea that antipsychotics can be preferable over SSRIs when treating sexually inappropriate behaviors in patients with pathologic irritability or mood instability given the potential worsening of these symptoms if SSRIs were to be used [5]. The authors were unable to find any clinical trials comparing the efficacy of antipsychotics and SSRIs for the management of inappropriate sexual behaviors. However, the literature does state a preference for the second-generation antipsychotics risperidone or olanzapine are preferred over their first-generation counterparts like haloperidol due to better patient tolerance of side effects [5]. In addition to the anti-dopaminergic activity of risperidone, it also has the added benefit of having the highest potential for hyperprolactinemia, which could provide more anti-libidal activity compared to other antipsychotics [3,5].

Anti-androgen therapy has also been explored as an option for managing inappropriate sexual behaviors, especially regarding paraphilias such as pedophilia [64]. A randomized clinical trial found that degarelix, a gonadotropin-releasing hormone (GnRH) antagonist, significantly reduced the risk for committing child sexual abuse 2 weeks after the initial injection [64]. Among the 26 participants who received degarelix as part of the randomized clinical trial, 20 (77%) experienced positive effects on sexuality (improved attitude or behavior) and 23 (89%) reported adverse effects on the body (suicidal ideation, transient injection site reactions, and elevated hepatobiliary enzyme levels) [64]. Only 2 (8%) of the degarelix patients endorsed suicidal ideation and were subsequently hospitalized, neither acted upon these suicidal thoughts [64]. The majority of the adverse side effects reported were transient injection site reactions at 2 weeks with 22 of 25 (88%) endorsing this and elevated hepatobiliary enzyme levels with 11 of 25 (44%) endorsing this [64]. In a post hoc analysis, the rates of suicidality, depression, and depression severity did not differ between the degarelix and control (placebo) group between two and ten weeks [64].

In addition to GnRH antagonists, 5α-reductase inhibitors such as finasteride have been studied for managing inappropriate sexual behaviors [3]. Similarly to GnRH antagonists, the purpose of using 5α-reductase inhibitors is to inhibit libido, this is achievedI by inhibiting the formation of dihydrotestosterone, a potent androgen [3]. This can lead to sexual dysfunction, a beneficial side effect in this context, but the potential for PFS must also be considered [3,62,63,65]. The true prevalence of PFS is unclear, given underreporting, limited clinician knowledge of the syndrome, and the lack of high-quality clinical trials with long-term follow up [62]. PFS can have a significant impact on the mood, in terms of depression and anxiety, of patients affected by it and there are no guidelines for the treatment of PFS [62]. On the other hand, there have been clinical trials on the adverse side effects of second-generation anti-androgens when used for the treatment of prostate cancer [66]. An increased risk of cognitive toxic effects (RR, 2.10; 95% CI, 1.30–3.38; *p*  =  0.002), fatigue (RR, 1.34; 95% CI, 1.16–1.54; *p*  <  0.001), and risk of falls (RR, 1.87; 95% CI, 1.27–2.75; *p*  =  0.001) was noted among individuals treated with second-generation anti-androgens compared to those in the control arms [66]. This demonstrates how, despite its potential for effectively managing sexually inappropriate behaviors, there are still risks associated with anti-androgens which differ from the other drug classes a clinician can use. There is little evidence to suggest that one class (SSRIs, antipsychotics, or anti-androgens) is superior to the other in managing inappropriate sexual behavior, instead the decision of which drug class to use must be made based on the individual circumstances of a patient [3].

### 3.5. Strength of Existing Literature

The off-label use of SSRIs to manage inappropriate sexual behaviors in psychiatric patients is a growing area of clinical interest, though the strength of supporting evidence remains variable. While randomized controlled trials would constitute the gold standard for establishing efficacy, much of the existing support derives from case reports, case series, theoretical papers, preclinical animal studies, and observational studies, making the evidence base more suggestive than definitive [25,33].

Clinical trials directly evaluating SSRIs for the explicit management of sexual impulsivity or paraphilic behavior are limited in number and by the specificity of their participant populations [25,33]. For instance, there have been clinical trials that investigate SSRI use among incarcerated individuals convicted of sexual offenses [25,33]. These studies have found that SSRI use was associated with a marked reduction in sexual preoccupation, fantasy, and risk-related thoughts [25,33]. 

On the other hand, there are comparatively more mechanistic studies that examine the potential of SSRIs for this use [15]. Studies have explored how SSRIs and SNRIs influence sexual side effects through modulation of dopamine and serotonin, providing insight into why these agents might reduce hypersexual behavior [15]. Though not clinical trials, such pharmacological analyses reinforce the rationale for using SSRIs in managing sexual dysregulation [15].

Case reports and survey studies have also contributed to the literature, with some demonstrating how SSRIs and finasteride can impact sexual function [47,48]. While the focus of these studies was on adverse effects, these findings indirectly support the notion that SSRIs can modulate sexual behaviors powerfully, though the lasting nature of these changes raises concerns about reversibility and safety [47,48]. These concerns about reversibility and safety were indirectly addressed in a 2024 study that evaluated “drug holidays” in patients experiencing SSRI-induced sexual dysfunction [31]. The reversibility of side effects during these breaks offers indirect evidence of SSRIs’ dose-dependent impact on sexual function, further validating their potential for behavioral modulation when appropriately managed [31].

Upon assessing the risk of biases, the authors found that the case reports and case series were generally of high quality with little to no concerns of biased reporting. Similarly, the cohort/observational studies included as references had little to no concerns about biased reporting, though approximately half of the references did not assess participants multiple times and none of the outcome assessors appeared to be blinded to the exposure status of participants. There were also no concerns about biased reporting from any of the animal studies included as references, but the majority did not specify how the animals were housed during the experiment, whether animals were randomly selected for the experiment, and none of the animal studies stated if caregivers/investigators or outcome assessors were blinded from knowledge of the intervention during the experiment. All narrative and systematic reviews included as references were of high quality, properly referencing sources for important statements but most did not go into detail regarding how their literature searches were conducted. A summary of the references can be seen in Table 2 below.

Overall, the evidence supporting SSRI use for managing sexual behaviors in psychiatric patients is largely indirect but growing. While mechanistic and observational studies offer compelling rationale and clinical feasibility, the lack of large-scale, placebo-controlled trials limits definitive conclusions. Nonetheless, the consistent finding across the studies shows that SSRIs frequently suppress various aspects of sexual function and provide a strong foundation for continued clinical application and future research. 

### 3.6. Prevalence of Sexual Side Effects and Treatment Options for Them

SSRIs are frequently reported in psychiatric literature for their sexual side effects, but their specific use in managing sexually inappropriate or compulsive behaviors is a less frequently studied application [25,35,37,38]. Most of the literature addressing this use case is observational or inferential, rather than based on large-scale clinical trials [38]. The most direct evidence comes from a 2024 clinical trial which studied SSRI use in incarcerated individuals with histories of sexual offenses [25]. The authors reported that SSRIs were used to reduce sexual preoccupation and intrusive sexual thoughts in forensic settings [25]. 

Case reports and reviews have similarly described the suppressive sexual side effects of SSRIs as potentially useful in treating conditions like paraphilic disorders or hypersexuality [15,58]. These studies note that serotonergic medications are sometimes used off-label in clinical psychiatry for patients exhibiting inappropriate sexual behaviors, particularly in inpatient or forensic populations [15,58]. However, these instances are discussed more in terms of theoretical or case-based utility than as part of a systematic clinical practice [15,58].

The prevalence of sexual side effects associated with SSRI use is well documented and consistent across a range of populations [2,11,22,48]. One study found that the prevalence of sexual dysfunction is highly prevalent among both females (88.7%) and males (84.5%) taking psychotropic medications [11]. This study noted that prior literature has reported rates of 70% to 80% of patients will experience sexual dysfunction while on any psychotropic medications, but it was found that SSRIs and SNRIs potentially have even higher rates of sexual dysfunction in patients, with up to 93% of participants reporting at least one sexual dysfunction [11]. Therefore, the prevalence of having some form of sexual dysfunction secondary to an SSRI or SNRI appears rather high, ranging between 84.5% to 93% as reported in the literature [2,11,22,48].

Studies examining PSSD have found that its prevalence is not well-understood or documented but estimate that its prevalence is potentially between 0.46% and 26.3% of SSRI users [13,22,29]. The 0.46% prevalence value was determined through a retrospective cohort study carried out using the medical database of the Cialit Hospital System in Tel Aviv by identifying the number of phosphodiesterase 5 inhibitor prescriptions in males 20–50 without other comorbidities who had stopped SSRI treatment at least one month prior to the data collection [13]. As there are no guidelines for the treatment of PSSD, the use of phosphodiesterase 5 inhibitors as a measure to quantify the prevalence of PSSD could underestimate the true prevalence of the disorder which is something the authors of the study acknowledged [13]. On the other hand, the 26.3% prevalence rate was based on an online-survey study conducted of 76 participants composed of former antidepressant users who were surveyed for evidence of persisting sexual effects [14]. Of this small sample, 20 (26.3%) participants suffered from genital anesthesia and/or nipple insensitivity which is suggestive of PSSD [14]. As demonstrated by the varied approach in identifying potential cases of PSSD, there is a lack of standardized diagnostic criteria for the condition which can result in under or over-reporting of the incidence of PSSD [13,14,15]. Symptoms of PSSD that appear to be generally agreed upon include genital numbness, absence of libido, and orgasmic difficulties [22]. The findings of these prevalence studies point to a concerning, long-term consequence of SSRI therapy that clinicians must consider when using SSRIs or SNRIs for managing sexual behaviors in inpatient psychiatric units [22]. However, future studies with more comprehensive and standardized diagnostic criteria are needed before the true prevalence of PSSD can be adequately estimated. For example, a multi-center randomized control trial with multiple SSRIs and a placebo arm for the management of inappropriate sexual behaviors in hospitals could provide data on the safety and efficacy of off-label SSRI use for inappropriate sexual behaviors. This, alongside longitudinal follow-up, for at least one year, could also provide data on the prevalence of PSSD following the use of SSRIs for the management of inappropriate sexual behaviors in inpatient settings, while also gathering a database of individuals with confirmed PSSD who may be recruited into future clinical trials for PSSD treatment. Once more data can be obtained regarding the frequency of PSSD through randomized control trials or database collection, additional information on the etiology and biochemistry of PSSD could be collected through the comparison of hormonal levels, genetic data, and neuroimaging between PSSD sufferers and those who receive comparable SSRI treatment without developing PSSD.

Although no guidelines from major organizations such as the American Psychiatric Association exist for the treatment of PSSD or for sexual dysfunctions occurring during SSRI/SNRI use, there have been studies that offer potential options, such as the concept of drug holidays, switching antidepressants, downward titration of the offending agent, add-on therapy, and watchful waiting [2,31]. To conduct a drug holiday for a patient, patients can skip their doses of SSRIs/SNRIs for a weekend in order to lower their serum levels of the medication in hopes that their sexual symptoms improve [31]. This was found to significantly improve erection, ejaculation, satisfaction, and overall sexual health in participants with no major side effects or changes in mental status being noted [31]. Of course, this should be carefully considered by clinicians and discussed with patients, depending on the patient’s past mental health history and reactions to psychotropic medications. Additionally, this study had a relatively small sample size; a total of 63 patients were included, so further studies are needed for more concrete and generalizable conclusions to be made [31]. 

On the other hand, a case report found that SSRIs that are seen as more activating to mitigate sexual side effects better than less activating SSRIs [37]. This potentially indicates that switching SSRI class or augmenting with other psychotropic agents, like Vortioxetine, is a viable method for addressing sexual dysfunction in SSRI patients [37,42]. However, there are no clinical trials which have explored this method in depth, so clinicians who do wish to switch the SSRI with another SSRI or to switch to a different antidepressant class will need to do so based on a theoretical, mechanistic basis [2]. In the meantime, patients whose SSRI-induced sexual side effects are treated by medication switching face the potential for a relapse of mood symptoms as the new antidepressant may not be as effective or may present with new side effects that were not present with the previous antidepressant [2]. 

Downward titration is another treatment option for SSRI-induced sexual dysfunction as prior literature has noted that there appears to be a dose-dependent relationship between the occurrence of sexual dysfunction and SSRIs [2,11,22]. However, some patients may exhibit sexual dysfunction at even partial doses due to pharmacodynamic and genetic factors [2]. There is also the concern that a reduction in the dosage of the SSRI can result in withdrawal symptoms or in a relapse of mood symptoms [2].

Add-on therapy appears to be better studied in the literature with some open label clinical trials reporting positive results associated with augmenting SSRI therapy with other medications, either to allow for a lowering of the dose of the offending SSRI or to directly manage the sexual dysfunction symptoms themselves [2]. The addition of bupropion, mirtazapine, and buspirone to an SSRI regimen was noted to improve sexual functioning in open-label studies according to a review [2]. The management of sexual symptoms by non-psychotropic medications like phosphodiesterase 5 inhibitors were found to be effective in a large number of trials in a review also for both men and women [2]. This option does have its downsides such as potentially increased healthcare costs for the patient, previously unstudied drug–drug interactions if unusual combinations of medications are used, and the add-on medication may have its own side effects the patient may need to cope with [2].

The last proposed treatment method by the literature is watchful waiting [2]. There is the possibility that the sexual dysfunction will resolve on its own or, if the SSRI regimen is a short-term or otherwise temporary one, patients may opt to endure the side effects until they finish their course [2]. However, studies have shown that 42% of men and 15% of women become noncompliant with SSRIs due to reported sexual side effects [2]. This significant noncompliance rate must be taken into consideration if a clinician were to choose this option. However, noncompliance of psychotropic medications can be mitigated through patient education [67]. A study found that clinical pharmacists who educated patients prior to discharge on the importance of medication compliance in order to avoid symptom exacerbation and rehospitalization led to statistically increased rates of medication compliance six to eight weeks post-discharge (*p* < 0.001) [67].

Overall, SSRIs are commonly used in psychiatry, and their sexual side effects are among the most consistently reported adverse reactions, affecting more than half of users [11]. However, their purposeful use for managing sexual behaviors in psychiatric patients is less frequently documented and supported primarily by observational studies and theoretical rationale, resulting in a lack of formal consensus or clinical algorithms for their use in this context. Concrete prevalence numbers for side effects like decreased libido and delayed orgasm are widely available, but the use of SSRIs to treat inappropriate sexual behaviors lacks both quantitative frequency data and endorsement from major psychiatric organizations. As such, this remains an area of clinical practice guided more by empirical judgment than formalized standards.

### 3.7. Use in Special Populations

SSRIs are most commonly used to manage sexual behaviors in adult psychiatric and forensic populations with relatively more data on the sexual effects of SSRIs being reported on these populations than pediatric patients which have much more limited but emerging studies beginning to be reported [25,32,33,57]. The bulk of clinical literature on vulnerable or special populations focuses on incarcerated or institutionalized adults, particularly individuals with a history of sexually inappropriate or compulsive behaviors [25]. One study reported on the use of SSRIs among individuals incarcerated for sexual offenses in the UK, noting that SSRIs were employed to reduce sexual ideation and preoccupation in this population [25]. The authors emphasized the utility of SSRIs in forensic psychiatric settings where behavioral control is a therapeutic priority, though noted that if more intensive libido suppression was required then anti-androgens would serve better than SSRIs [25]. The use of SSRIs in forensic populations has also been explored for the management of impulsive and aggressive behaviors with the theory that low serotonergic activity results in poor behavioral inhibition and an increased risk of violence [32,33]. A small 2010 clinical trial of 34 individuals found that of the 20 who completed a 3-month trial of sertraline were less impulsive, irritable, angry, depressed, and were also less likely to assault, verbally assault, or indirectly assault others [32]. All 20 participants who completed the 3-month trial of sertraline had requested to continue the treatment under the supervision of their own medical practitioners, potentially indicating that SSRIs can prove to be beneficial to forensic populations for a variety of reasons, not just for inappropriate sexual behaviors [32]. However, this is not an endorsement of the off-label use of SSRIs in forensic populations, further clinical trials with more concrete findings are needed to establish treatment guidelines for the use of SSRIs in forensic populations.

Pediatric use is less common and more controversial, though it is discussed in the context of long-term consequences rather than primary treatment [57]. SSRI exposure during adolescence was found to impact adult sexual behaviors, with some gender-specific differences being noted [57]. The study found that women with a history of SSRI use during development exhibited significantly lower solitary sexual desire, decreased attraction-based desire, and reduced masturbation frequency, while males showed little to no negative sexual side effects [57]. However, males with a history of non-SSRI were found to report higher levels of partnered sex activity in adulthood, but this is likely due to the small sample size of 3 male participants who reported non-SSRI treatment before the age of 16 [57]. The author of this study noted that the decrease in solitary sexual desire in women with a history of pediatric SSRI use was particularly concerning, given that solitary sexual desire is thought to more closely reflect the physiologic mechanisms sexual reward which is separate from a desire for intimacy or emotional closeness [57]. Exposure to serotonergic medications during development alters the function of the ventral palladium, raphe nucleus, amygdala, and ventromedial hypothalamus, areas important for sexual motivation and reward in women [57]. This is not reflected in individuals with treatment histories with non-SSRIs, indicating that the serotonergic activity of SSRIs is the cause of these changes as opposed to a confounding variable or mechanism of action [57]. SSRIs appear to have a stronger effect on the anticipation of sexual activity, decreasing one’s interest in sexual activity rather than decreasing one’s pleasure experienced during sexual stimulation if exposed during childhood development [57]. These findings indicate that SSRIs may have lasting sexual and neurodevelopmental effects if used during critical periods, suggesting that their use in pediatric populations should be approached with caution [57].

The prevalence of inappropriate sexual behaviors in geriatric patients with dementia reportedly ranges from 7% to 25%, with potentially higher rates in those who are residents of skilled nursing facilities [3]. Currently, the use of SSRIs for the management of inappropriate sexual behaviors in geriatric patients is not FDA approved, despite commonly being used as first-line agents due to their superior safety and tolerability profile in the elderly [3]. This is in comparison to antipsychotics which have black box labels warning against their use in geriatric patients with dementia as there is an increased risk of stroke and cardiac events in this population [3]. SSRIs such as paroxetine and citalopram were found to be effective in some case reports, according to a review, but these findings are inconsistent [3]. In the case of a 69-year-old male with alcoholic dementia and inappropriate sexual behaviors who had not responded to haloperidol, chlorpromazine, lorazepam, lithium, and nortriptyline, the patient responded positively within one week of starting paroxetine 20 mg [3]. Meanwhile, a 90-year-old woman had failed to respond to paroxetine but experienced a sharp reduction in aggressive and sexually disinhibited behavior when started on citalopram 20 mg [3]. No clinical trials have explored SSRI use in geriatric populations for sexually inappropriate behaviors, so clinicians will have to weigh the relative risks and benefits based on their own subjective judgement. However, a review indicated that the most common side effects other than sexual dysfunction in geriatric SSRI use are gastrointestinal problems, headache, and hypersensitivity reactions [3]. Additionally, there is also the risk of serotonin syndrome if an SSRI is prescribed with other serotonergic agents [3]. Careful titration and discontinuation of SSRIs in this population can help with avoiding severe adverse effects or with preventing withdrawal symptoms from occurring [3].

Overall, SSRIs are primarily used for managing sexual behaviors in adult and forensic psychiatric populations [25]. SSRIs have been shown to improve aggressiveness and reduce sexual ideation as well as sexual preoccupation in forensic populations [25,32]. There are currently ongoing clinical trials examining the use of SSRIs in forensic populations, so more conclusive results will ideally be available to clinicians in the near future [33]. Until then, clinicians will need to make decisions based on data on the side effects of SSRIs based on its role as an antidepressant as opposed to its off-label use as a pharmaceutical method for managing inappropriate sexual behaviors. Geriatric populations also have limited clinical data regarding the use of SSRIs for managing inappropriate sexual behaviors, but it should be noted that SSRIs are perceived as having superior safety profiles compared to antipsychotics in geriatric populations with dementia [3]. There are black box FDA warnings against the use of first and second-generation antipsychotics in geriatric patients with dementia [3]. The existing case reports for geriatric patients with dementia started on SSRIs are contradictory at times, with some SSRIs exhibiting clinical effectiveness within a week, decreasing patient sexual inappropriateness as well as aggression, while failing to achieve significant results in other geriatric patients with dementia [3]. Although SSRIs appear to have superior safety profiles compared to antipsychotics in geriatric populations, the lack of high-quality clinical trials limits any general recommendations or conclusions that can be made with the existing data which primarily consists of case reports. Therefore, the use of SSRIs in geriatric populations will also need to be decided on a case-by-case basis by clinicians based on the particular needs and reactions of the patient. Finally, the use of SSRIs in pediatric populations appears to affect the long-term sexual motivation of patients when they become adults, but do not appear to have an impact on the pleasure adults with a pediatric history of SSRI-use experience when undergoing sexual acts with a partner [57]. Further research is needed into pediatric SSRI use, but caution should be taken in the meantime given the potential for long-term developmental changes that have been reportedly associated with pediatric SSRI use [57].

## 4. Discussion

There are many things to take into account when considering the use of SSRI for managing inappropriate sexual behaviors in psychiatric patients. This includes careful weighing of the risks and benefits of this medication class in order to adhere to the principles of beneficence and non-maleficence. The incidence of SSRI induced sexual dysfunction is noted to be quite high in the literature, with up to 93% of participants reporting at least one sexual dysfunction [11]. However, there are no known clinical trials which examine the use of SSRI induced sexual dysfunction for managing inappropriate sexual behaviors, thus preventing concrete conclusions from being drawn for the formation of clinical recommendations or treatment guidelines. Instead, clinicians must carefully weigh multiple factors to see if the benefit of prescribing an SSRI to prevent inappropriate sexual behaviors that may affect other patients, hospital staff, or visitors to the psychiatric ward outweighs the potential risks that the SSRI poses for the patient in question [1].

In terms of its use as an antidepressant, SSRIs are known for having superior safety profiles compared to older generations of antidepressants such as monoamine-oxidase inhibitors and tricyclic antidepressants [3,4,5]. However, rare and potentially fatal side effects like hyponatremia and arrhythmias secondary to SSRI-induced QT-interval prolongation can occur, so the drug class is not devoid of dangers [9]. Additionally, the risk of PSSD is currently inadequately measured in the literature with significantly varied incidence rates proposed, ranging from 0.46% to 26.3% [13,14]. This is because PSSD is not well-known by clinicians and researchers have no standardized criteria for diagnosing the condition, leading to potential under and over-reporting of its prevalence [13,14]. This is further complicated by the lack of treatment guidelines for PSSD, meaning the condition can be an enduring and distressing unintended side effect for some starting on SSRIs. High-quality clinical trials will be needed in order to understand the prevalence, mechanisms, and treatment options for PSSD. 

This presents an issue for adhering to the principle of non-maleficence, as PSSD and similar disorders like PFS, can pose a significant risk to patients as they have been found to be associated with higher rates of depression and anxiety [62]. On the other hand, the more acute SSRI-induced sexual dysfunctions are thought to be more reversible than PSSD with five main methods that clinicians use such as drug holidays, switching antidepressants, downward titration of the offending agent, add-on therapy, and watchful waiting [2,31]. All of these treatment methods have their own benefits and cons as discussed previously. Given the limited information and lack of clinical trials for the prevalence and treatment of PSSD, a careful analysis of the potential benefits of the off-label use of SSRIs is warranted as the risks of PSSD are currently uncertain.

Between April to June of 2017, nearly 60,000 reports were filed with England’s National Health Service trust mental health wards and 1120 (1.6% of reports) were sexual incidents involving patients, staff, and visitors [1]. About two-thirds (594) of the people affected were categorized as patients and one third (301) were staff, with a small number (24) who were others, such as visitors to the ward [1]. One benefit of SSRIs, that has been proven in small clinical trials, is their ability to manage impulsive and aggressive behaviors through increased serotonergic tone [3,31,32,33,34]. Another proposed benefit is the relatively high prevalence of sexual dysfunction for patients prescribed SSRIs which can be harnessed therapeutically when prescribed off-label for sexually inappropriate psychiatric patients [11]. Depending on the severity of the sexually inappropriate behaviors, those around the patient may be put in substantial danger which could lead to the benefits of prescribing SSRIs off-label to outweigh the risk of PSSD. However, this is highly situational and there is not enough data to provide general guidelines for how to approach these scenarios.

The literature also discusses the off-label use of other drug classes, mainly antipsychotics and anti-androgens, for managing sexually inappropriate behaviors too. In comparison to these other drug classes, SSRIs are thought to have a superior safety profile, especially when it comes to off-label uses in geriatric patients with dementia [3]. There are currently black box warnings from the FDA against the use of antipsychotics in geriatric patients with dementia due to increased risks of stroke and cardiac events in this population [3]. Meanwhile, anti-androgen treatment options are controversial for a separate reason. They are seen as, “chemical castration,” and are stigmatized by the public when used off-label for managing sexually inappropriate behaviors [3]. Given the social stigma against anti-androgens and the super safety profile compared to antipsychotics, SSRIs can be considered a more acceptable option for managing inappropriate sexual behaviors in geriatric populations [3]. However, the literature on this topic is also limited mainly to case reports, so identifying which specific SSRIs are superior in terms of managing sexually inappropriate behaviors from clinical trial data is not possible. 

Instead, a review found that some studies have shown that SSRIs and serotonin norepinephrine reuptake inhibitors (SNRIs) are associated with higher rates of sexual dysfunction (citalopram, fluoxetine, paroxetine, sertraline, and venlafaxine) as compared to placebo, whereas newer antidepressants such as amineptine, agomelatine, bupropion, moclobemide, nefazodone, and mirtazapine were associated with a lower incidence of sexual dysfunction [2]. The remaining antidepressants show an intermediate risk of sexual dysfunction, and this subset includes fluvoxamine, escitalopram, duloxetine, and imipramine [2]. Therefore, if a clinician wants to prescribe SSRIs off-label to manage inappropriate sexual behaviors, then citalopram, fluoxetine, paroxetine, sertraline, and venlafaxine are some of the first antidepressants that they should consider [2]. 

Outside of clinical decision making, there is also the ethical issue of ensuring proper informed consent and obtaining it for the off-label use of SSRIs. While clinicians may be able to inform patients about the safety of SSRIs, the variance in the reported prevalence and potentially enduring adverse side effect that is PSSD can prove to be an issue for obtaining informed consent. In patients with dementia who lack capacity to make informed decisions, it may be possible to obtain informed consent for medications through a surrogate decision maker. However, patients who have poor insight but maintain decision-making capacity may refuse to take SSRIs as they may believe that their inappropriate sexual behaviors are acceptable, such as in a patient experiencing an acute manic episode. Being worried about developing PSSD is a valid concern, persistent sexual dysfunction induced by a medication can have significant consequences on one’s mental health [62]. Therefore, non-pharmaceutical options such as patient education, changing the patient’s environment, and psychotherapy should be exhausted before attempting pharmaceutical treatment [1]. If all of these attempts fail and the patient continues to refuse psychotropic medications, then the next steps depend on the local or regional laws regarding involuntary psychotropic medication administration. If emergency intramuscular medications are required because a patient is imminently likely to harm someone and is also exhibiting inappropriate sexual behaviors, then an antipsychotic is likely the best option out of the three discussed in this review since antipsychotics can be sedating, exhibit anti-libidinal effects, and are available as injectables unlike SSRIs [3]. 

Ideally, a patient exhibiting inappropriate sexual behaviors will also have another comorbid condition which would be suitably treated by one of the pharmaceutical options discussed in this review. In individuals with major depressive disorder or anxiety, SSRIs would be ideal while those with schizophrenia spectrum disorders or those with a history of mood instability/mania would benefit more from antipsychotics [5]. Regardless of patient psychiatric history, it is important to properly educate patients through structured clinical support, ideally with the assistance of clinical pharmacists who have been shown to improve adherence to complex pharmacotherapies, including those with adverse drug reactions such as sexual dysfunction [67]. Approximately 42% of men and 15% of women are noncompliant with SSRIs due to sexual dysfunction, so monitoring of sexual dysfunction severity is important to ensure continued outpatient compliance with this off-label use of SSRIs [2]. There are various ways to track the severity of sexual dysfunction, such as a thorough initial patient history and continued follow-up with questions about sexual dysfunction [2]. However, if a numerical value is preferred for long-term trending, then there are various measures used in the literature such as the male sexual health questionnaire and female sexual function index [28,31].

### Limitations

This review is mainly limited by the lack of clinical trials which examine the off-label use of SSRIs for the management of inappropriate sexual behaviors. The majority of the literature discussing the efficacy and side effects of SSRIs for this off-label use are predominantly case reports, case series, and observational studies. Although the case reports included in this review were found to be of high-quality, reporting many of the necessary details regarding patient history, demographics, and interventions, patient responses to SSRIs were inconsistent at times. A 2016 review found a case report of a 69-year old male had his inappropriate sexual behaviors improve within a week of starting paroxetine 20 mg, but this same review also found a case report of a 90-year old woman who failed to respond to paroxetine and instead experienced a sharp reduction in aggressive and sexually inappropriate behaviors when started on citalopram [3]. This further highlights the pressing need for high-quality clinical trials which can control for confounding variables, be conducted with substantially larger sample sizes, and be used to better assess the prevalence of PSSD.

Additionally, the criteria for diagnosing disorders like PSSD and PFS are currently not standardized. These disorders are not widely recognized and have not been included in the Diagnostic and Statistical Manual of Mental Disorders. In fact, there are claims in the literature that PFS is a delusional disorder, however biological changes have been found in individuals with PFS which indicate it is a genuine disorder [47]. This general lack of awareness of these conditions and varying diagnostic criteria cause significant variance in their reported prevalences. PFS does not appear to have an estimated prevalence, but the prevalence of PSSD ranges from 0.46% to 26.3% in the literature, a difference of over 5717% [13,14]. Official treatment guidelines also do not exist for these conditions, so any potential treatment options put forth by this review are mainly informed by either theoretical references or by general trends in case reports; both of these types of sources lack the statistical strength and validity of randomized controlled trials.

## 5. Conclusions

The literature widely reports the high prevalence of patients taking SSRIs experiencing at least one form of sexual dysfunction, which indicates that the off-label use of SSRIs for managing sexual behaviors will likely cause some form of sexual dysfunction to occur in patients. Although normally considered a common adverse effect of SSRIs, this SSRI-induced sexual dysfunction has the potential to be used therapeutically for the safety of others in an inpatient psychiatric setting. This can prove to be useful, especially for geriatric patients with dementia, as SSRIs have a superior safety profile compared to antipsychotics, which are also used off-label for managing sexually inappropriate behaviors but have FDA black box warnings placed on them for their use in geriatric patients with early dementia. However, the prevalence of some significant adverse effects that can persist despite the discontinuation of SSRIs, like PSSD, are still poorly understood and can therefore pose an issue when obtaining informed consent from patients. Being concerned about developing PSSD is a valid concern that an average individual would likely be worried about; therefore, the management of sexually inappropriate behaviors should be taken in a stepwise fashion.

First, clinicians should exhaust all non-pharmaceutical options such as educating the patient on why their inappropriate sexual behaviors are not acceptable, changing the patient’s environment to improve the safety of those around the patient, or using cognitive behavioral therapy. If these options have all failed or the patient is an imminent danger to others, then a clinician should consider pharmaceutical options. SSRIs have the benefit of having relatively superior safety profiles, the ability to treat mood disorders, and being less costly than many other pharmaceuticals unless it is one of the fewer SSRIs. However, a patient will need to take SSRIs orally, which allows for patients to avoid taking their medications. Antipsychotics on the other hand, are available as intramuscular injectable forms, which allows for emergency administration and long-term maintenance therapy through a long-acting injectable. Depending on the generation of antipsychotic, the side effects can differ such as extrapyramidal symptoms being more common in first-generation antipsychotics, while second-generation antipsychotics present with more metabolic side effects. As mentioned previously, antipsychotics are also generally avoided in geriatric patients with dementia given the increased risk of stroke and cardiac events on their black box warnings. Finally, there are the anti-androgens such as the 5α-reductase inhibitors like finasteride and the GnRH antagonists like degarelix, which have been shown to be effective in treating paraphilias, like pedophilia, in forensic populations. However, the use of anti-androgens has a social stigma attached to it, which patients and their families may be concerned about. Some believe that anti-androgens are, “chemical castration,” which violates an individual’s rights. Additionally, PFS is also poorly understood and the discussion of its potential during informed consent may cause patients or their families to be hesitant to provide their consent for anti-androgens.

Ideally, patients exhibiting inappropriate sexual behaviors will have a pre-existing psychiatric or medical condition which can be treated with one of these main drug classes. In the case that this occurs, clinicians are recommended to take a thorough initial history which includes questions about prior sexual dysfunction, current sexual habits, and potentially the use of clinical measures like a male sexual health questionnaire or female sexual function index if one wishes to track a quantifiable score over time [28,31]. By maintaining open communication about sexual side effects from these medications and potentially including a clinical pharmacist to further assist in educating patients, clinicians can improve patient medical compliance through therapeutic use of drug-induced sexual side effects while also not allowing said side effects to become too severe. There are multiple treatment options for drug-induced sexual dysfunction, though none of these are officially advocated for in organization guidelines. These treatment options include drug holidays, switching antidepressants, downward titration of the offending agent, add-on therapy, and watchful waiting. Each has their own pros and cons, as discussed above, so a combination of clinical judgement from the healthcare provider and the opinions/values of the patient should dictate which treatment options should be explored first. Overall, clinicians should be aware that the literature regarding the off-label use of SSRIs for managing inappropriate sexual behaviors is primarily based on case reports and preclinical animal models, therefore lacking the solid clinical evidence that would be associated with high-quality randomized controlled trials. Until such clinical trials exist, clinicians should remain cautious when using SSRIs for off-label purposes and carefully weigh the risks and benefits as discussed in this review.

## Figures and Tables

**Figure 1 healthcare-13-02433-f001:**
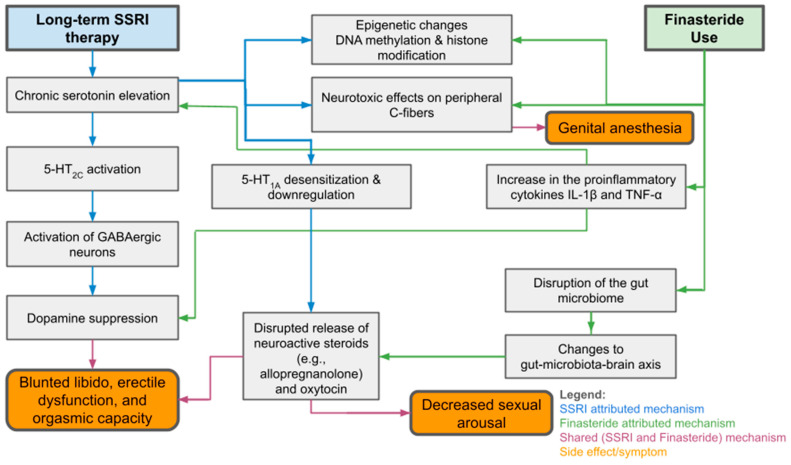
Overview of Mechanisms and Sexual Side Effects of SSRI and Finasteride Use.

**Table 1 healthcare-13-02433-t001:** Odds Ratio of Antidepressant Sexual Dysfunction and PSSD Prevalence.

Reference	Medication Name	Drug Class	Odds Ratio and 95% Confidence Interval
Zeiss et al., 2024 [29]	Agomelatine		4.35 (2.07–9.14)
Zeiss et al., 2024 [29]	Bupropion		5.83 (5.01–6.8)
Zeiss et al., 2024 [29]	Citalopram	SSRI	13.88 (12.19–15.81)
Zeiss et al., 2024 [29]	Clomipramine		4.94 (3.25–7.5)
Zeiss et al., 2024 [29]	Duloxetine	SNRI	11.08 (9.9–12.41)
Zeiss et al., 2024 [29]	Escitalopram	SSRI	20.59 (18.32–23.14)
Zeiss et al., 2024 [29]	Fluoxetine	SSRI	7.56 (6.68–8.56)
Zeiss et al., 2024 [29]	Fluvoxamine	SSRI	4.88 (1.65–11.75)
Zeiss et al., 2024 [29]	Milnacipran	SNRI	4.7 (2.44–9.05)
Zeiss et al., 2024 [29]	Mirtazapine	Tetracyclic antidepressant	3.95 (2.98–5.25)
Zeiss et al., 2024 [29]	Paroxetine	SSRI	21.78 (20.1–23.59)
Zeiss et al., 2024 [29]	Reboxetine	SNRI	14.77 (8.89–24.55)
Zeiss et al., 2024 [29]	Sertraline	SSRI	16.94 (15.5–18.5)
Zeiss et al., 2024 [29]	Trazodone	Serotonin antagonist and reuptake inhibitor	2.91 (1.99–4.24)
Zeiss et al., 2024 [29]	Venlafaxine	SNRI	8.96 (7.9–10.17)
Zeiss et al., 2024 [29]	Vortioxetine	Serotonin modulation and stimulator	20.84 (17.31–25.09)
Reference	Range of Reported Prevalence of PSSD	Prevalence of PSSD	Study Type
Ben-Sheetrit et al., 2023 [13]	-	0.46%	Retrospective Cohort Study
Healy et al., 2024 [14]	-	26.3%	Review of Animal Study
Pirani et al., 2024 [53]	-	13.2%	Observational Cohort Study
Waraich et al., 2020 [30]	-	4%	Retrospective Cohort Study
Overall Literature	0.46% to 26.3%	-	-

**Table 2 healthcare-13-02433-t002:** Reference Study Designs, Sample Sizes, and Main Outcomes.

Reference	Type	Sample Size	Main Outcomes/Notes
Esquivel et al., 2020 [55]	Animal study	Rats	An animal study examining the impact of the serotonin transporter (SERT) gene has on sexual behaviors. Selective pre- and postsynaptic 5-HT_1A_ receptor agonists possess pro-sexual effects in SERT+/+ and SERT−/−, although the response is diminished in SERT−/− animals, most likely due to desensitization of 5-HT_1A_ receptors.
Giatti et al., 2021 [40]	Animal study	Rats	SSRI treatment alters neuroactive steroid levels and the expression of key enzymes of steroidogenesis in a brain-tissue and time-dependent manner. The results of the animal study indicate that the effect of paroxetine treatment is directly on neurosteroidogenesis with a negative impact on the expression of steroidogenic enzymes observed on withdrawal of paroxetine treatment. The authors hypothesize that altered neurosteroidogenesis may occur in PSSD.
Giatti et al., 2024 [36]	Animal study	Rats	Using RNA, the transcriptomic profile of the hypothalamus and nucleus accumbens of male rats treated daily for 2 weeks with paroxetine (T0) and at 1 month of withdrawal (T1). 7 differentially expressed genes were found at T0 and 1 at T1 in the hypothalamus, 245 at T0 and 6 at T1 in the nucleus accumbens. Genes related to neurotransmitters with a role in sexual behavior and the ward system were found to be dysregulated in the nucleus accumbens, supporting the idea of dysfunction in this brain area. Analysis of differentially expressed genes at T1 in the nucleus accumbens confirmed the persistence of some side effects, providing more information on post-SSRI sexual dysfunction etiopathogenesis.
Santana et al., 2019 [12]	Animal study	Rats	Tyrosine hydroxylase immunoreactivity decreased significantly in the substantia nigra and ventral tegmental area after treatment with paroxetine and labeling was reduced significantly in the zona incerta and mediobasal hypothalamus. The immunoreactive axons in the striatum, cortex, hippocampus, and median eminence almost disappeared in paroxetine-treated rats. Treatment with agomelatine caused a moderate reduction in immunoreactivity in the substantia nigra without appreciable modifications to other regions of the brain. The authors conclude that paroxetine, but not agomelatine, is associated with important decreases in activity in dopaminergic areas of the brain associated with sexual performance impairment in humans after antidepressant treatment.
Dewan et al., 2012 [43]	Case Report	1 male	A 55-year-old male with PTSD and MDD initially treated with bupropion monotherapy which failed to adequately control his symptoms. Sertraline was added and this resulted in improved psychiatric symptoms, but also hypersexuality which gradually resolved one month after stopping sertraline.
Klaas et al., 2023 [15]	Case Report	1 male	A 55-year-old male was started on venlafaxine for depressive symptoms cause by burnout. Over a course of 5 years, the patient attempted to stop treatment twice after his mood improved but developed low libido, delayed ejaculation, erectile dysfunction, “brain zaps”, overactive bladder, and urinary inconsistency each time. After gradually weaning off of venlafaxine gradually over 1.5 months (75 mg to 35.5 mg to 0 mg), the patient was able to stop taking the medication. The patient subsequently experienced sexual dysfunction approximately a year after stopping his venlafaxine, indicating the patient developed PSSD. At the time of writing, the patient had been experiencing symptoms of PSSD but had not received a diagnosis previously.
Mahalakshmi et al., 2025 [35]	Case Report	1 female	A 25-year-old female taking 15 mg of escitalopram per day for depressive symptoms developed intense sexual desire and compulsive masturbation which vanished with discontinuation of the drug
Mania et al., 2006 [44]	Case Report	1 male	A 54-year-old male with bipolar disorder and cognitive deficits secondary to Parkinson disease had been exhibiting intermittent inappropriate sexual behavior for the past 5 years. At one point he was hospitalized, and all of his psychiatric medications were stopped except lithium, this resulted in worsening of his inappropriate sexual behavior. After failing behavioral therapies, the patient was started on 20 mg citalopram and within 2 weeks, his inappropriate sexual behaviors had resolved.
Moses et al., 2023 [37]	Case Report	1 male	A 36-year-old male patient with OCD and anxiety was initially prescribed paroxetine, then sertraline, and then fluvoxamine 250 mg. He also received buspirone 10 mg twice daily. The patient was taking 150 mg fluvoxamine for a year with limited effects on his OCD symptoms but was bothered by adverse effects like fatigue and diminished sex drive. The patient transitioned from fluvoxamine to fluoxetine 40 mg per day and continued on buspirone, 25 days later the patient was no longer complaining about fatigue nor sexual dysfunction.
Patacchini et al., 2020 [27]	Case Report	1 male	A 21-year-old male began treatment with sertraline 100 mg per day for a major depressive episode. During treatment, the patient was irritable, emotionally flat, and felt detached with no perceived benefits from the medication. Psychotherapy helped with his symptoms. After 2 years of treatment with sertraline, the patient was tapered on sertraline. The patient experienced premature ejaculation during the taper and experienced a sense of physical impotence, absence of libido, and genital anesthesia in addition to a recurrence of his mood symptoms. The patient continued to experience these symptoms for 4 years, during which time he was seen by psychiatrists, a specialist of sexual disorders, and an andrologist. Lab tests during this time were normal, but the andrologist noted a non-elicitable bulbocavernosus reflex and hypesthesia/dysesthesia of the genital area. The patient saw some improvement with pramipexole 1 mg per day, but this was continued due to unspecified side effects. The patient also experienced some benefits with bupropion at 300 mg per day for 2 days. Sertraline was reintroduced for 2 months without any benefits noted.
Yuan et al., 2022 [38]	Case Report	1 male	A 42-year-old male with a history of papillary thyroid carcinoma status post total thyroidectomy and left modified radical neck dissection with a full course of radioactive iodine treatment presented to a psychiatric clinic for persistent fatigue, increased anxiety, and irritability. A screening measure indicated a moderate level of depression, and his levothyroxine requirements were closely followed by his endocrinologist. The patient was started on bupropion XL 150 mg once daily, but this was discontinued due to increased irritability. Sertraline 50 mg once daily was started with improvements of his symptoms, but he also developed hypersexuality (including increased sex drive, constant sexual thoughts, and compulsive masturbation throughout the day) as well as delayed ejaculation. The patient was then trialed on duloxetine 30 mg one daily and then escitalopram but elected to restart sertraline 50 mg daily because he felt that the side effects outweighed the risks. No sexual symptoms were noted during this second trial of sertraline.
Stachelek et al., 2025 [46]	Case Series	3 males	Three male participants were seen in a urology clinic and treated with high frequency electrical stimulation and low intensity extracorporeal shock wave therapy for a total of 16 weeks for sexual dysfunction induced by SSRI or finasteride use. Mild to moderate improvements in sexual function was noted in all three participants.
Winder et al., 2024 [25]	Non-randomized, open label trial	135 males	A non-randomized, open label study of 77 incarcerated males in the United Kingdom with sexual convictions were treated with anti-androgens (*n* = 8) or SSRIs (*n* = 69). 66 received Fluoxetine, 2 Paroxetine, and 1 Sertraline. 58 participants were assigned to the control group for comparison. Both medicated groups demonstrated levels of problematic sexual arousal 3 months post-baseline, the comparison group did not.
Lorenz 2020 [57]	Observational Cohort Study	610 people (207 males, 403 females)	An observational survey study of 610 young adults assessed childhood and current mental health, detailing past antidepressant and other psychopharmaceutical prescription history before the age of 16. For women, childhood SSRI use was associated with significantly lower solitary sexual desire, desire for an attractive other, and frequency of masturbation. No differences in women’s partnered sexual desire or sexual activity was noted. Childhood use of non-SSRI antidepressants or non-antidepressant psychiatric medications were not associated with adult sexual desire/behavioral changes in either women or men.
Melcangi et al., 2017 [63]	Observational Cohort Study	16 males	Small observational cohort study of 16 young male patients aged 22–44 with post-finasteride syndrome. 50% screened positive for depression after developing PFS. All patients showed some form of erectile dysfunction with 10 (62.5%) men affected severely and 6 (37.5%) with mild-moderate forms. This cohort of patients showed a low score for orgasmic function, sexual desire and overall satisfaction domains, compared to the general population. The levels of some neuroactive steroids analysed in CSF of PFS patients were significantly different versus those in healthy controls. In particular, the levels of pregnenolone, as well as of its further metabolites, progesterone and dihydroprogesterone, were significantly decreased in CSF of PFS patients. On the contrary, the levels of DHEA and testosterone were significantly increased.
Safak et al., 2025 [11]	Observational Cohort Study	451 people (291 males, 161 females)	An observational survey study using the Psychotropic-related Sexual Dysfunction Questionnaire comprised 452 people (291 males and 161 females). Sexual dysfunction was highly prevalent among both females (88.7%) and males (84.5%). Significant differences were observed based on antidepressant type with bupropion users experiencing lower levels of sexual dysfunction compared to others using SSRIs, SNRIs, or vortioxetine. No significant differences were found for males.
Landgren et al., 2020 [64]	Randomized clinical trial	52 males	In a randomized clinical trial of 52 males with pedophilic disorder, treatment with degarelix was found to significantly reduce the risk for committing child sexual abuse 2 weeks after initial injection. This finding suggests that degarelix may serve as a rapid-onset, risk-reducing medication for men with pedophilic disorder. The main adverse reactions were local injection site reactions and elevated hepatobiliary enzyme levels. 2 (8%) of the participants in the degarelix group reported suicidal ideation, but post hoc analysis found no statistically significant differences between suicidality when comparing the degarelix and placebo group.
Alipour-Kivi et al., 2024 [31]	Randomized, open-label, controlled trial	50 males	A small randomized, open-label, controlled trial with 63 patients. 32 patients were assigned to drug holiday groups and 31 were assigned to the control group. 50 patients completed the trial (25 in each group). Drug holidays from SSRIs significantly improved erection, ejaculation, satisfaction, and the overall sexual health of participants. No significant changes were observed in the mental health of drug holiday participants.
Ben-Sheetrit et al., 2023 [13]	Retrospective Cohort Study	12,302 males	A 19-year retrospective cohort analysis was conducted by querying a local hospital database for chart information on males aged 21 to 49 with erectile dysfunction (defined as a prescription of phosphodiesterase 5 inhibitors). Those with erectile dysfunction were grouped based on whether they used serotonergic antidepressants or not. Serotonergic antidepressants were associated with significantly higher rates of erectile dysfunction. The prevalence of PSSD was estimated to be 0.46%.
Luca et al., 2022 [42]	Retrospective Cohort Study	13 males	A retrospective cohort study of 13 males aged 29.53 ± 4.57 years with PSSD were tentatively treated based on their symptoms using an antidepressant with a dopaminergic/noradrenergic profile or antagonizing/positively modulating the serotonergic system (i.e., with fewer or no known SD side effects) as well as nutraceuticals and/or PDE5 inhibitors. Vortioxetine was the most commonly used and effective treatment out of all medications used.
Waraich et al., 2020 [30]	Retrospective Cohort Study	43 males	A retrospective chart review from 2009 to 2019 was performed finding 43 male patients who met the criteria for PSSD, approximately 4% of the male patients seen during that timeframe. 40 (93%) of these male patients had erectile dysfunction. Consistent with other reports of PSSD, the patients observed to have PSSD are young (mean age 31) with severe erectile dysfunction affecting most patients.
Lalegani et al., 2023 [28]	Randomized, open-label, controlled trial	50 females	A small randomized, open-label, controlled trial consisted of 55 female participants (drug holiday group n = 28, control group n = 27) examining the effects of drug holidays on female sexual function and mental health status. A total of 50 participants completed the trial (25 per group), the drug holiday group experienced significant improvements in arousal, desire, orgasm, satisfaction, lubrication, and overall sexual health.
Zeiss et al., 2024 [29]	Retrospective Pharmacovigilance Study	91,195 safety report cases	A study examining the World Health Organization’s database of individual case safety reports. A total number of 91,195 cases were examined from the database regarding desire, arousal, orgasm, and sexual dysfunction. Using this data, the authors were able to calculate odds ratios with their respective 95% confidence intervals for desire, arousal, orgasm, and sexual dysfunction. Serotonergic agents such as SSRIs and SNRIs were among the most frequently reported group of antidepressants linked to sexual dysfunction, with SSRIs such as paroxetine having higher reporting odd ratios (RORs) than TCAs such as clomipramine (Paroxetine ROR: 21.78 (20.1–23.59), clomipramine ROR: 4.94 (3.25–7.5))
Pirani et al., 2024 [53]	Observational Cohort Study	2179 survey participants	A survey study with 2179 participants which examined US/Canada sexual and gender minority youth aged 15 to 29. 574 (26.3%) of participants reported genital hypoesthesia, these participants were generally older and more likely to report their assigned sex at birth being male, having undergone hormonal therapy previously, and have a psychiatric drug history. The frequency of persistent post-treatment genital hypoesthesia among antidepressant users was 13.2% (93/707) compared to 0.9% (1/102) among users of other medications. An adjusted odds ratio was 14.2.
Butler et al., 2010 [32]	Non-randomized, open label trial	20 males	An open label clinical trial examining the effectiveness of a three-month trial of sertraline on controlling impulsive violent behaviors in individuals with a history of violent offending (at least one prior conviction for a violent offense). 34 individuals started the trial with 20 completing the three-month intervention, reductions in impulsivity, irritability, anger, assault, verbal-assault, indirect-assault, and depression were noted. All 20 participants who completed the three-month trial requested to continue sertraline under the supervision of their own medical practitioner. The findings of this study suggest that treating impulsive, violent individuals in a forensic setting with SSRIs can be beneficial for patient mood and to attenuate violent behaviors.
Patacchini et al., 2020 [27]	Observational Cohort Study	135 participants	An online survey study which examined 135 participants (115 males, average age 31.9 with a SD of 8.9 years) for self-declared PSSD. The survey found that PSSD was more common in younger, heterosexual males after exposure to SSRI/SNRI at relatively high doses. 118 subjects had symptoms both during and after SSRI/SNRI administration while only 17 participants experienced symptoms of PSSD after SSRI/SNRI treatment. The authors concluded that PSSD is a complex iatrogenic syndrome in need for further study.
Studt et al., 2021 [49]	Observational Cohort Study	239 participants	A survey study conducted through online support groups for individuals with PSSD, 239 survey responses were recorded. The majority of respondents reported a history of SSRI use (92%) compared to only SNRI or atypical antidepressant use (8%). The severity of symptoms improved in 45% of respondents and worsened/remained the same for 37% of respondents after discontinuing treatment with SSRIs. Only 12% of respondents reported receiving counseling regarding potential sexual dysfunction while taking antidepressants. Most rated the effect of PSSD on their quality of life as severely negative (59%) or very negative (23%).
Rice et al., 2025 [51]	Observational Cohort Study	10 participants	10 participants were recruited through a patient advocacy group to participate in private, semi-structured interviews. 8 main themes emerged upon phenomenological analysis to describe the participants’ experiences with PSSD. Individuals with PSSD undergo psychological, physical, and sexual effects of withdrawal that cause suffering, hopelessness, and alienation. The current lack of understanding, awareness, and informed consent or acceptance among healthcare providers about PSSD worsens the negative experiences associated with PSSD and contributes to an overall lack of trust in physicians and/or medicine in general.
Issa et al., 2025 [9]	Retrospective Cohort Study	234,217 participants	A retrospective study based on the Stockholm Sodium Cohort data of 1,632,249 individuals. First-time users of SSRI/venlafaxine were included, comprising 234,217 participants of whom 39,999 developed profound hyponatremia at least once. The incidence of profound hyponatremia among individuals 65 to 79 years old and greater than 80 years old were 3% and 4%, respectively. The odds ratio for profound hyponatremia was 4.29 (95% CI: 3.34–5.52) within the first 3 months after SSRI/venlafaxine initiation. After 1 year the aOR was 1.30 (95% CI: 0.97–1.75). During the first 2 weeks after initiation treatment, the aOR was 10.06 (95% CI: 5.97–17.00).

## Data Availability

No new data was created or analyzed in this review. Data sharing is not applicable to this review.

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
