# Peer review of "The Off-Label Use of Selective Serotonin Reuptake Inhibitors for Sexual Behavior Management: Risks and Considerations"

_healthcare, 2025, doi:10.3390/healthcare13192433_

Round 1

Reviewer 1 Report

Comments and Suggestions for Authors

This manuscript explores the off-label use of selective serotonin reuptake inhibitors (SSRIs) for managing inappropriate sexual behaviors in psychiatric patients. Its main strength lies in its discussion of a clinically relevant yet rarely addressed therapeutic application. It integrates neurobiological mechanisms, risks such as post-SSRI sexual dysfunction (PSSD), and ethical dilemmas. The manuscript also uses up-to-date literature and acknowledges the need for further research.

  1. The introduction jumps between several topics, such as pharmacology, history, epidemiology, and adverse effects, without establishing a clear, logical progression. I recommend improving this section.
  2. Correct the nomenclature of the 5-HT2C receptor, writing 2C in subscript. Make the same correction for all serotonergic receptors mentioned in the text and in Figure 1.
  3. The methodology description is limited. Even in a narrative review, it is essential to describe how the sources were selected in a structured way to avoid inclusion bias. There is no mention of how many articles were ultimately selected or how many were initially identified. The inclusion and exclusion criteria are also not indicated.
  4. Add tables or comparative graphs of prevalence by sex, type of SSRI, duration of treatment, etc.
  5. Summarizing the findings with figures would make the document more appealing to readers.
  6. The conclusion states that using SSRIs to treat inappropriate sexual behavior is "probably very effective," but this statement lacks solid clinical evidence or randomized controlled trials. I suggest that the authors exercise more caution.
  7. Include a section on the limitations and perspectives of the work.
  8. Adhere to the reference guide for formatting all references. Also, review the format of references 3 and 24.

Reviewer 2 Report

Comments and Suggestions for Authors

I read with interest the paper titled "The Off-Label Use of Selective Serotonin Reuptake Inhibitors for Sexual Behavior Management: Risks and Considerations."

1. Methods section is underdeveloped. A clearer description of search strategy, selection criteria, and article screening process is needed to enhance transparency and reproducibility.

2. The abstract could be improved with better structure and clarity. For instance, some sentences are overly long and repetitive.

3. The paper alternates between the terms "sexual disinhibition," "hypersexuality," and "sexually inappropriate behaviors." It would benefit from consistent terminology and clearer definitions early in the Introduction.

4. The Conclusion section reiterates many points but lacks a strong, action-oriented final message. 

Reviewer 3 Report

Comments and Suggestions for Authors

General assessment

The manuscript entitled “The Off-Label Use of Selective Serotonin Reuptake Inhibitors (SSRIs) for Sexual Behavior Management: Risks and Considerations” provides a narrative synthesis on the potential use of SSRIs to manage hypersexuality and inappropriate sexual behaviors in psychiatric patients. The topic is clinically important, given the absence of standardized guidelines for this indication, and the rising interest in balancing therapeutic suppression of libido against the risk of persistent sexual dysfunction. The review is clearly structured and cites both mechanistic studies and case-based clinical evidence. However, several aspects need strengthening before this article can be recommended for publication.

Major strengths

Clinical relevance: The paper addresses a practical problem frequently encountered in psychiatric and geriatric inpatient units, where inappropriate sexual behaviors may compromise care and safety.

Integration of mechanistic and clinical data: The authors describe serotonergic modulation of dopaminergic reward circuits, as well as case reports and small series, providing a useful framework to understand how SSRIs might reduce sexual impulsivity.

Attention to post-SSRI sexual dysfunction (PSSD): The recognition of PSSD as a serious and underdiagnosed long-term adverse effect is timely, and the manuscript usefully highlights ethical considerations.

Major concerns and recommendations

Methodological transparency: The search strategy (Google Scholar and PubMed) is too vague. The authors should specify keywords, date ranges, inclusion/exclusion criteria, and whether any bias assessment was undertaken. This is essential to ensure reproducibility.

Evidence quality and grading: The manuscript cites multiple case reports, observational studies, and mechanistic papers. However, it does not differentiate clearly between levels of evidence. A table summarizing study designs, populations, sample sizes, and main outcomes would help readers contextualize the strength of evidence.

Risk–benefit balance: While the review underscores risks such as PSSD, it sometimes overstates efficacy based on limited or anecdotal data. The clinical benefits of SSRIs in this context should be framed more cautiously, especially since no randomized controlled trials exist.

Clinical applicability: The paper should provide practical guidance for clinicians. For example, which SSRIs are more potent in libido suppression? How should clinicians monitor for emergent sexual dysfunction? What ethical frameworks should guide off-label prescribing in vulnerable populations?

Special populations: The review briefly mentions pediatric and geriatric patients, but more detailed analysis is warranted. Adolescent exposure to SSRIs and its potential long-term sexual consequences is especially concerning and requires a stronger cautionary statement.

An article (PMID: 31343497) emphasizes the importance of structured clinical support and pharmacist involvement in improving adherence to complex pharmacotherapies with ADRs including sexual dysfunction. Recommending it here would broaden the scope of the review by underlining that beyond efficacy and safety of SSRIs, long-term outcomes also depend on medication adherence, monitoring, and multidisciplinary management.

Suggested insertion for the Discussion section:

“Given that adherence challenges are common in psychiatric populations, integrating clinical pharmacists into care teams may help ensure safe off-label prescribing and monitoring of SSRIs. Evidence from schizophrenia spectrum disorders shows that pharmacist involvement can significantly improve compliance and treatment outcomes.”

A good place to integrate an article (PMID: 23192413) is where the manuscript discusses tolerability issues with SSRIs, especially insomnia, anxiety, and sexual dysfunction. The authors emphasize that off-label use of SSRIs for behavioral control must balance risks such as persistent sexual dysfunction against clinical benefits. At that point, highlighting trazodone’s distinct pharmacological profile (SARI mechanism) would provide readers with a relevant comparator.

Suggested insertion for the Discussion section:

“It is also worth noting that alternative serotonergic agents such as trazodone, a serotonin receptor antagonist and reuptake inhibitor (SARI), may address some of the tolerability limitations frequently observed with SSRIs. Clinical studies demonstrate that trazodone has comparable antidepressant efficacy to SSRIs and SNRIs while potentially mitigating insomnia, anxiety, and sexual dysfunction. The development of prolonged-release once-daily formulations has further improved its tolerability and adherence profile. Although primarily approved for major depressive disorder, low-dose trazodone is also commonly used off-label as a hypnotic, which underscores the broader relevance of antidepressant selection when considering off-label strategies for behavioral management.”

Minor comments:

Ensure consistent terminology (e.g., “sexual disinhibition,” “hypersexuality,” “compulsive sexual behaviors”).

Standardize prevalence figures: in some places, 70–80% prevalence of SSRI-induced sexual dysfunction is cited, while elsewhere up to 93% is reported. Provide consistent ranges with references.

Clarify whether percentages are reported per patient, per treatment episode, or per sexual symptom.

Conclusion and recommendation:

Overall, the manuscript addresses a clinically relevant and underexplored topic. However, the current version requires major revision to strengthen methodological rigor, clarify the hierarchy of evidence, and provide more practical clinical guidance.

Reviewer 4 Report

Comments and Suggestions for Authors

This article is an exceptionally well referenced, well written, and thorough review of the sexual side effects of selective serotonin reuptake inhibitors used for sexual behavior management.  It is a needed contribution to the literture.  I have no other specific criticisms.  The authors are to be congratulated on their review.

Reviewer 5 Report

Comments and Suggestions for Authors

Title of the Manuscript: The Off-Label Use of Selective Serotonin Reuptake Inhibitors for Sexual Behavior Management: Risks and Considerations

General Assessment

This narrative review addresses an important and clinically relevant issue: off-label use of SSRIs for the treatment of hypersexuality and premature sexual behavior in psychiatric patients. The authors offer an exhaustive integration of current knowledge, highlight the therapeutic potential of SSRIs, and indicate the associated risks, namely post-SSRI sexual dysfunction (PSSD). The topic is timely, given increased awareness of PSSD and ethical concerns regarding the long-term adverse effects of psychotropic medication.

The manuscript is overall well written and understandable, but also has certain drawbacks that need to be overcome before it gets published.

Methodological limitations

The review relies exclusively on Google Scholar and PubMed searches without a detailed search strategy or PRISMA-style framework. This weakens the transparency and reproducibility of the review process.

Exclusion/inclusion criteria are poorly described. For example, whether relevant but non-English studies were excluded, and whether grey literature was considered, is not stated.

The evidence base is heavily weighted towards small observation studies and case reports. While the authors acknowledge this, the manuscript tends to overestimate the strength of conclusions at times. The need for randomized controlled trials needs to be emphasized more.

The manuscript provides extensive detail on the risks (especially PSSD) but gives less weight to the benefits and clinical contexts where SSRIs may be clearly preferable over alternatives such as anti-androgens. Expanding this discussion would provide a more balanced perspective.

The paradoxical hypersexuality part is interesting but not adequately fleshed out. It could benefit from a sharper theoretical explanation and exploration of potential mechanisms.

Specific Suggestions

Expand the discussion on ethical implications of prescribing SSRIs off-label for sexual behavior management, particularly regarding informed consent and monitoring.

Recommendation

Major Revision

The review has the potential to be of high contribution to the literature but requires methodological clarification, greater balance in risk-benefit consideration, and less conservative interpretation of findings. Addressing these issues will make the academic rigor and clinical relevance of the manuscript stronger.

Reviewer 6 Report

Comments and Suggestions for Authors

The publication entitled “The Off-Label Use of Selective Serotonin Reuptake Inhibitors for Sexual Behavior Management: Risks and Considerations” examines a significant and clinically pertinent issue at the convergence of psychiatry, pharmacology, and ethics. The authors are praised for highlighting the often-overlooked topic of off-label SSRI usage in the management of sexual behavior, especially among psychiatric populations. The review is prompt and emphasizes both therapeutic advantages and notable concerns, including post-SSRI sexual dysfunction (PSSD).

However, the manuscript requires major revisions before it can be considered for publication.

Abstract:

The abstract must be better organized, concisely capturing the context, objectives, methodologies (databases consulted, inclusion criteria), principal findings, and consequences. At present, it resembles an introduction. Authors must explicitly delineate the ethical dilemmas and clinical deficiencies within the guidelines.

Introduction:

  1. Kindly include epidemiological data regarding inappropriate sexual behaviors in populations with dementia and psychiatric disorders.
  2. The integration of neurobiological hypotheses, such as epigenetics and receptor desensitization, is essential for understanding the persistence of PSSD.
  3. The historical section on SSRI approval could be condensed; instead, emphasize clinical relevance.
  4. A comparison of SSRIs and anti-androgen therapy regarding their efficacy and safety is advisable.
  5. The authors delve into the topic of SSRIs, primarily focusing on their application in PTSD treatment. However, there is a noticeable lack of discussion regarding the effects of sertraline and paroxetine. Consequently, I suggest incorporating additional information, for instance.

https://pubmed.ncbi.nlm.nih.gov/40493868/

https://pubmed.ncbi.nlm.nih.gov/38185385/

  1. An exploration of the variations in off-label SSRI use across different countries and healthcare systems is needed. It would be beneficial to examine the factors influencing these differences, including regulatory frameworks, prescribing practices, and cultural attitudes towards mental health treatment. A comprehensive analysis could shed light on the implications for patient care and the overall effectiveness of SSRIs in diverse contexts.
  2. It is essential to provide clear definitions for the terms hypersexuality, sexual disinhibition, and paraphilic disorders, as each represents a unique concept that warrants distinct clarification.
  3. It is essential to delve into the reasons behind the absence of established guidelines, particularly in light of the prevalent off-label use. What factors contribute to this gap in regulation and standardization? An exploration of the underlying issues is warranted to understand the implications of such a discrepancy.
  4. The stigma surrounding the reporting of sexual dysfunction in psychiatric patients is a significant barrier to effective treatment and open communication between patients and healthcare providers. Many individuals may feel embarrassed or ashamed to discuss these issues, fearing judgment or misunderstanding. This reluctance can lead to underreporting of symptoms, ultimately hindering the ability to address and manage sexual health concerns adequately. Addressing this stigma is crucial for fostering a supportive environment where patients feel safe to discuss their experiences and receive the care they need.

Methods:

  1. The authors need to specify if they adhered to PRISMA guidelines or employed other systematic approaches in their methodology.
  2. The article would benefit from a clearer presentation of the inclusion and exclusion criteria, specifically regarding the years covered and the populations included.
  3. It would be beneficial to provide a clear account of the total number of studies that were screened and ultimately included in the analysis.

Results:

  1. The section discussing proposed mechanisms would benefit from the inclusion of schematic figures illustrating the interaction between serotonergic and dopaminergic systems.
  2. It is essential to incorporate endocrine considerations, particularly regarding HPG axis suppression, supported by robust clinical evidence.
  3. This article would benefit from elaborating on receptor-level effects, particularly in light of recent preclinical studies involving 5-HT1A agonists.
  4. It would be beneficial to incorporate a comparative table that outlines the sexual side-effect profiles of various SSRIs.
  5. The literature section should prioritize the ranking of evidence, placing randomized controlled trials above cohort studies, followed by case reports.
  6. It is recommended to enhance the subsection on “special populations” by providing a detailed summary of evidence related to geriatric, forensic, and pediatric groups, each presented distinctly.

Discussion:

  1. The authors are encouraged to enhance the ethical discourse surrounding informed consent, risk communication, and monitoring.
  2. It is essential to consider various alternative strategies, including behavioral therapies, anti-androgens, and GnRH analogues. Each of these approaches offers unique benefits and potential outcomes that warrant thorough examination.
  3. It is necessary to elucidate the manner in which clinicians ought to balance the immediate behavioral advantages against the potential long-term sexual repercussions.
  4. Therapeutic interventions in the field of antidepressant treatment can be quite complex. One common approach is the switching of SSRIs, which may be necessary when a patient does not respond adequately to their initial medication. Additionally, augmentation strategies, such as the introduction of bupropion or vortioxetine, can be considered to enhance the efficacy of the primary treatment. These methods reflect a tailored approach to managing depression, aiming to optimize patient outcomes through careful medication management.
  5. It is essential to identify the existing gaps in the current body of research, particularly emphasizing the need for large randomized controlled trials and the exploration of biomarkers that could indicate the risk of PSSD.

Round 2

Reviewer 1 Report

Comments and Suggestions for Authors

Correct the nomenclature of the serotonergic receptors listed in Table 2. The accepted way to indicate serotonergic receptor subtypes is to use subscript letters (see https://www.guidetopharmacology.org/GRAC/GPCRListForward?class=A).

Author Response

Correct the nomenclature of the serotonergic receptors listed in Table 2. The accepted way to indicate serotonergic receptor subtypes is to use subscript letters (see https://www.guidetopharmacology.org/GRAC/GPCRListForward?class=A).

  • Thank you for noticing this. We have updated the serotonergic receptors listed in Table 2

Reviewer 3 Report

Comments and Suggestions for Authors

Thank you for submitting the revised version by taking my suggestions into account.

Author Response

Thank you for submitting the revised version by taking my suggestions into account.

  • Thank you for your suggestions

Reviewer 5 Report

Comments and Suggestions for Authors

The authors successfully respond to all my concerns.

Author Response

The authors successfully respond to all my concerns

  • Thank you for your suggestions

Reviewer 6 Report

Comments and Suggestions for Authors

Initially, I would like to commend the authors for their meticulous response to the comments. The abstract has been reorganized appropriately, the introduction now incorporates epidemiological data and more precise terminology, the methodology is more detailed, and the results include comparative tables and mechanistic insights. Additionally, the ethical discourse has been considerably expanded, which enhances the overall composition of the manuscript.

However, a few minor points remain that, if addressed, will further improve clarity and impact:

1] Ensure consistent terminology between the abstract and text (e.g., “inappropriate sexual behavior” vs. “sexual disinhibition”) all throughout the manuscript.

2] The figure illustrating serotonergic–dopaminergic interaction is clear, but adding a short explanatory legend for non-specialist readers would increase accessibility.

3] In some places, “PSSD” is expanded fully, while in others, only the acronym is used. Please maintain consistency.

4] While the manuscript now explains 5-HT1A and 5-HT2C receptor roles, I would recommend expanding the discussion on neuroendocrine aspects, particularly HPG axis suppression and prolactin changes, with more recent human or translational data.

5] The revised manuscript emphasizes the need for randomized controlled trials, but it could be strengthened by outlining specific research strategies, such as biomarker development (genetic, neuroimaging, endocrine markers) to predict susceptibility to PSSD.

6] Ensure uniform formatting and adherence to journal guidelines (remove PubMed links).

Author Response

Initially, I would like to commend the authors for their meticulous response to the comments. The abstract has been reorganized appropriately, the introduction now incorporates epidemiological data and more precise terminology, the methodology is more detailed, and the results include comparative tables and mechanistic insights. Additionally, the ethical discourse has been considerably expanded, which enhances the overall composition of the manuscript.

However, a few minor points remain that, if addressed, will further improve clarity and impact:

1] Ensure consistent terminology between the abstract and text (e.g., “inappropriate sexual behavior” vs. “sexual disinhibition”) all throughout the manuscript.

  • Thank you for noticing this, we have checked our terminology and updated sexual disinhibition to inappropriate sexual behaviors in 3 cases. The remaining use of the words, “sexual disinhibition” and “hypersexuality” are appropriate as they are used in the definition of the phrases or are used to reflect the language of case reports

2] The figure illustrating serotonergic–dopaminergic interaction is clear, but adding a short explanatory legend for non-specialist readers would increase accessibility.

  • We have added a legend that should aid readers in understanding the mechanism steps. We also updated the figure as the initial revision of the figure did not include some of the specific end-symptoms that we had included in the first version of the figure.

3] In some places, “PSSD” is expanded fully, while in others, only the acronym is used. Please maintain consistency.

  • Thank you for noticing this, we have checked the manuscript and replaced all cases of post-SSRI sexual dysfunction with PSSD other than the initial case where the abbreviation is initially shown alongside the full phrase.

4] While the manuscript now explains 5-HT1A and 5-HT2C receptor roles, I would recommend expanding the discussion on neuroendocrine aspects, particularly HPG axis suppression and prolactin changes, with more recent human or translational data.

  • The authors attempted to find human clinical trials examining the interactions of SSRIs with the HPG axis, but were unable to find publicly available data. We note this and mention one example of a clinical trial examining the effects of antidepressants on sex hormones and sexual functioning (NCT00611975) which was completed, but never publicly published its data/results. Instead, we opted to turn to the results of animal studies to examine the effects of SSRIs on enzymes involved in steroidogenesis.

5] The revised manuscript emphasizes the need for randomized controlled trials, but it could be strengthened by outlining specific research strategies, such as biomarker development (genetic, neuroimaging, endocrine markers) to predict susceptibility to PSSD.

  • We have included a short statement that recommends future clinical trials to be multi-centered randomized control trials with longitudinal follow-up (at least 1 year) to examine both the efficacy/safety of SSRI use in managing sexual behaviors and the prevalence of PSSD after this off-label use. We also note a benefit of this study being the creation of a potential registry of confirmed PSSD patients who may benefit from other future potential randomized control trials that would examine PSSD treatment protocols.

6] Ensure uniform formatting and adherence to journal guidelines (remove PubMed links).

  • The authors were unable to find any PubMed links in our reference section. We do include the DOI’s of the references. We believe that leaving the DOI’s for the copyeditors will assist them in formatting the references in the future, so we will leave them in for now. We will remove the DOI’s and follow journal formatting guidelines during the production stage if this manuscript is accepted. Thank you